# Genome editing in animals with minimal PAM CRISPR-Cas9 enzymes

Jeremy Vicencio [1,11], Carlos Sánchez-Bolaños[2,3,11], Ismael Moreno-Sánchez [2,3], David Brena[1], Charles E. Vejnar [4], Dmytro Kukhtar[1], Miguel Ruiz-López[1], Mariona Cots-Ponjoan[1], Alejandro Rubio[2,3], Natalia Rodrigo Melero[5], Jesús Crespo-Cuadrado[2], Carlo Carolis [5], Antonio J. Pérez-Pulido [2,3], Antonio J. Giráldez [4,6,7], Benjamin P. Kleinstiver [8,9,10], Julián Cerón [1✉] & Miguel A. Moreno-Mateos [2,3✉]

The requirement for Cas nucleases to recognize a specific PAM is a major restriction for genome editing. SpCas9 variants SpG and SpRY, recognizing NGN and NRN PAMs, respectively, have contributed to increase the number of editable genomic sites in cell cultures and plants. However, their use has not been demonstrated in animals. Here we study the nuclease activity of SpG and SpRY by targeting 40 sites in zebrafish and *C. elegans*. Delivered as mRNA-gRNA or ribonucleoprotein (RNP) complexes, SpG and SpRY were able to induce mutations in vivo, albeit at a lower rate than SpCas9 in equivalent formulations. This lower activity was overcome by optimizing mRNA-gRNA or RNP concentration, leading to mutagenesis at regions inaccessible to SpCas9. We also found that the CRISPRscan algorithm could help to predict SpG and SpRY targets with high activity in vivo. Finally, we applied SpG and SpRY to generate knock-ins by homology-directed repair. Altogether, our results expand the CRISPR-Cas targeting genomic landscape in animals.

[1] Modeling human diseases in C. elegans Group; Genes, Disease and Therapy Program, Institut d'Investigació Biomèdica de Bellvitge - IDIBELL, L'Hospitalet de Llobregat, 08908 Barcelona, Spain. [2] Andalusian Center for Developmental Biology (CABD), Pablo de Olavide University/CSIC/Junta de Andalucía, Ctra. Utrera Km.1, 41013 Seville, Spain. [3] Department of Molecular Biology and Biochemical Engineering, Pablo de Olavide University, Ctra. Utrera Km.1, 41013 Seville, Spain. [4] Department of Genetics, Yale University School of Medicine, New Haven, CT 06510, USA. [5] Biomolecular Screening and Protein Technologies Unit, Centre for Genomic Regulation (CRG), The Barcelona Institute of Science and Technology, Barcelona 08003, Spain. [6] Yale Stem Cell Center, Yale University School of Medicine, New Haven, CT 06510, USA. [7] Yale Cancer Center, Yale University School of Medicine, New Haven, CT 06510, USA. [8] Center for Genomic Medicine, Massachusetts General Hospital, Boston, MA 02114, USA. [9] Department of Pathology, Massachusetts General Hospital, Boston, MA 02114, USA. [10] Department of Pathology, Harvard Medical School, Boston, MA 02115, USA. [11]These authors contributed equally: Jeremy Vicencio, Carlos Sánchez-Bolaños. ✉email: jceron@idibell.cat; mamormat@upo.es

CRISPR-Cas technology has become a revolution in molecular biology, biotechnology, and biomedicine[1]. However, the requirement for a Cas-specific protospacer adjacent motif (PAM) prohibits targeting many genomic regions. The inability to target freely within a genome limits the selection of the most optimal targets for precision genome editing. This is particularly critical when targeting and editing short regions in the genome such as miRNA loci or targets, transcription factor binding sites, enhancers, or promoters. The distance from the cut site or PAM to the edit is also essential for generating the desired mutation by homology direct repair (HDR)[2] or base editing, respectively[3]. To alleviate this handicap, newly discovered or engineered Cas proteins have increased the number of targets with different or more flexible PAMs[4–11]. In particular, SpG and SpRY are two modified versions of SpCas9 with more relaxed PAM requirements than NGG and, consequently, can target a greater fraction of the genome (Supplementary Fig. 1a, b). Specifically, SpG showed high activity in targets with NGN PAMs whereas SpRY could target nearly every sequence in the genome, with targets harboring NRN (R = A or G) PAMs being more efficiently edited than those with NYN (Y = C or T) PAMs[6]. However, most of the new or modified Cas nucleases, including SpG and SpRY, are first optimized and examined in mammalian cell culture without validation in animal models.

Here, we characterize and optimize SpG and SpRY in two different animal models widely used in biology and biomedical research: zebrafish (Danio rerio) and nematodes (Caenorhabditis elegans). We show that purified proteins or mRNAs coding for SpG or SpRY under optimized conditions determined in this study can target and edit DNA at different loci upon injection in zebrafish embryos and C. elegans germlines[12]. In addition, we show that the in vivo prediction algorithm CRISPRscan[13] can discriminate between highly active and inefficient gRNAs for SpG and SpRY, facilitating the use of these nucleases in vivo. Finally, we provide a C. elegans strain that endogenously expresses SpG, thus simplifying genome editing in this animal model. Altogether, our results expand the CRISPR-Cas toolbox in vertebrate and invertebrate animals and set a baseline for the use of minimal PAM engineered SpCas9 variants in other animal models.

## Results

**Optimization of SpG and SpRY for editing zebrafish and C. elegans genomes.** We assessed the activities of SpG and SpRY in animals by testing them in two distinct model organisms, zebrafish and C. elegans. To compare the activity of different nucleases in vivo, we injected one-cell stage zebrafish embryos using mRNAs coding for SpCas9 (WT), SpG, and SpRY targeting two sites with NGG PAMs in the gene slc45a2 (albino), previously analyzed for SpCas9[13] (Fig. 1a). The lack of function of slc45a2 is observed as a loss or reduction in pigmentation that can be quantified (Fig. 1b–d)[13]. While we were able to recapitulate the phenotypes induced from one high-efficiency gRNA and another that generates lower lack of pigmentation[13], experiments with SpG and SpRY revealed inferior levels of editing on NGG PAM at standard concentrations for gRNAs and Cas9 mRNA (Fig. 1c), similar to as previously reported in human cells[6]. We hypothesized that this lower activity with SpG and SpRY in our experiments might result from suboptimal concentrations of gRNA and Cas9 mRNA injected in zebrafish embryos. Thus, by increasing the concentration of mRNA (300 pg per embryo) and/or gRNA (240 pg per embryo), we observed a significant enhancement in activity as evidenced by higher penetrant phenotypes for both gRNAs (Fig. 1d). Such increase did not significantly affect embryonic viability, suggesting a lack of toxicity-associated effects (Fig. 1d, Supplementary Fig. 2a–c). Interestingly, increasing the

amount of mRNA and gRNA per embryo also improved SpCas9 activity but to a lesser extent than SpG and SpRY for the more efficient gRNA (Supplementary Fig. 2b, c). However, much higher concentrations of mRNA and gRNA significantly decreased embryo viability but did not significantly enhance SpG and SpRY activity, suggesting that an excess of these two components does not improve the system's performance (Supplementary Fig. 2a, d, e).

In parallel, we tested SpG and SpRY in C. elegans via RNP delivery since it has been shown to be more efficient than plasmid-based delivery and it limits the period of Cas activity[14,15]. We purified His tagged wild-type SpCas9, SpG, and SpRY for performing in vitro and in vivo experiments (Fig. 1e). Since the activity of Cas9 orthologs found in diverse microbes varies with temperature[7], we examined whether the activity of the engineered SpG and SpRY variants displayed temperature sensitivity that would hamper their use in other organisms growing at temperatures below 37 °C such as zebrafish and C. elegans. We tested these proteins in vitro at different temperatures: 15 °C, 25 °C, 37 °C, and 50 °C. All three purified proteins, along with a commercially obtained wild-type Cas9 protein, generated targeted DNA double-strand breaks on a substrate harboring an NGG PAM with similar efficiency (Supplementary Fig. 3a). To analyze the specificity of these proteins, we investigated their sensitivity to mismatches located in the protospacer sequence. Remarkably, SpG and SpRY were similarly sensitive to mismatches proximal to the PAM (position +1 and +5) at 25 °C or 37 °C (Fig. 1f). Then, since mutant phenotypes correlate well with CRISPR activity in C. elegans[16] (Supplementary Fig. 3b–e), we validated the tolerance to mismatches in vivo by scoring a dominant phenotype caused by dpy-10 targeting, by using SpG with the matched gRNA and a gRNA with a mismatch at position +5 (Fig. 1g). In a separate experiment, we compared the activities of SpG and SpRY with dpy-10 gRNAs, both matched and mismatched at the +5 position (from Fig. 1f) by scoring phenotypes (Supplementary Fig. 4a) and quantifying mutagenesis by ICE (Interference of CRISPR Editing, Synthego, Supplementary Fig. 4b). We observed that a single mismatch almost completely abolished the in vivo activity of SpG and SpRY. Therefore, the amino acid substitutions in these variants do not lead to relaxed recognition of the protospacer sequence relative to SpCas9, which is important for maintaining specificity. Then, we compared Cas9, SpG, and SpRY activities in vivo at two different sites (dpy-10 and gtbp-1::wrmScarlet) with NGG PAMs.

Dominant mutations of the C. elegans gene dpy-10 produce an overt phenotype (dumpy or roller animals) in the $F_1$ generation that we use for scoring CRISPR-Cas activity. In particular, there is an efficient NGG gRNA for dpy-10 that is commonly used as a co-CRISPR control for effective microinjections[17,18]. However, other dpy-10 alleles are recessive, producing a phenotype in the $F_2$ generation instead (Supplementary Fig. 4c). We first used the standard gRNA and ssDNA repair template (to generate the cn64 allele) for dpy-10 as in co-CRISPR assays and found that SpCas9 was more efficient than SpG and SpRY in producing dpy-10 mutations against sites with an NGG PAM (Supplementary Fig. 4d). Since the half-life of these nucleases is limited, we scored their efficiency at two different periods and observed that it was lower after 24 h (Supplementary Fig. 4d).

We targeted an additional locus to confirm the gradient of efficiency SpCas9 > SpG > SpRY on sites with NGG PAMs. By scoring the absence of fluorescence in the $F_2$ of animals with the endogenous reporter gtbp-1::wrmScarlet, we evaluated the efficiency of these three nucleases (Fig. 1h). This experiment corroborated the gradient of activity among the distinct Cas nucleases targeting an NGG PAM, with SpCas9 again

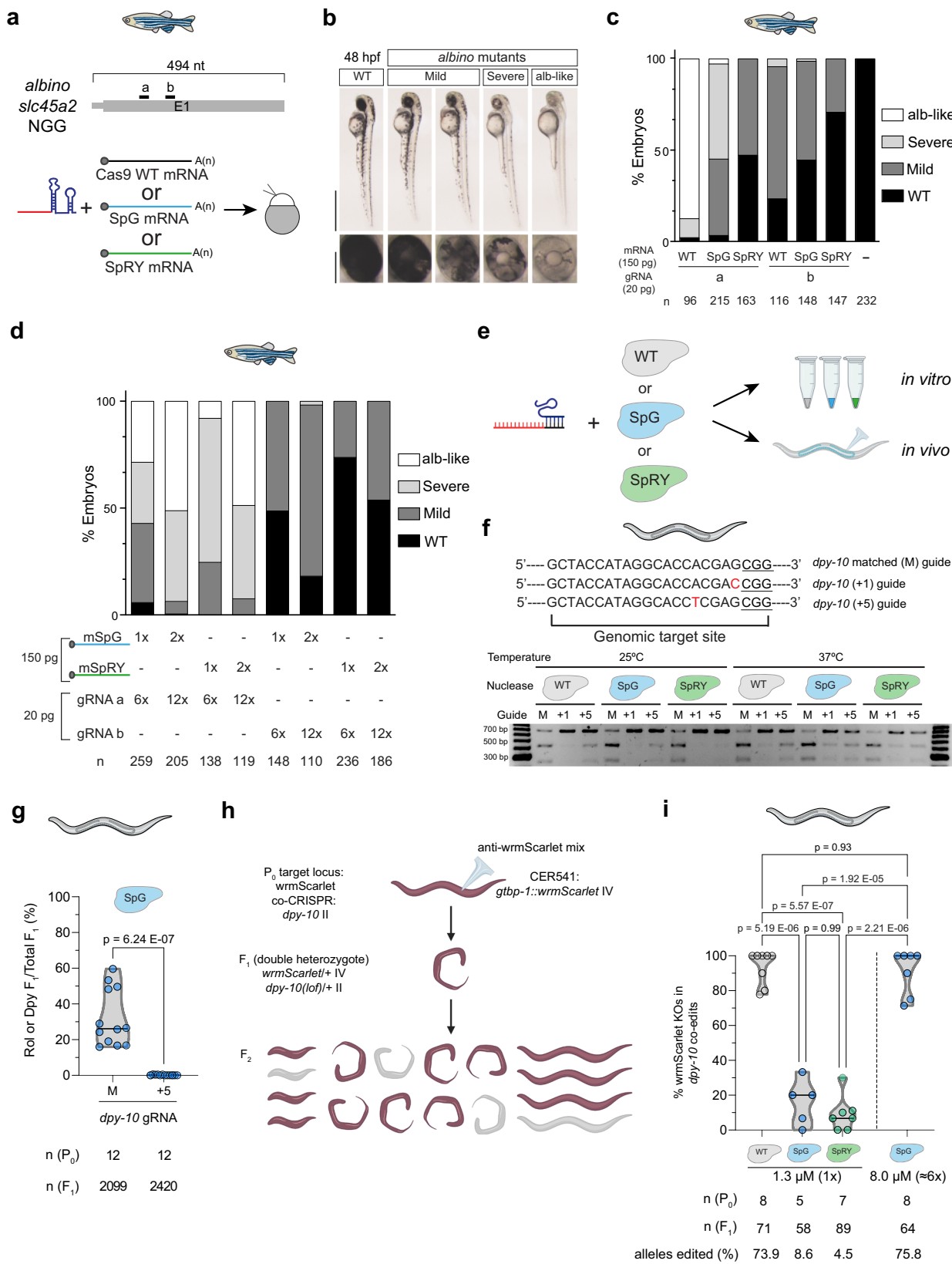

having the highest efficiency, and SpRY the least (Fig. 1i). To control microinjection quality, we also used the *dpy-10* co-CRISPR strategy.

In a previous study, we found that Cas9 concentrations can be raised six-fold without any apparent toxicity[19]. Thus, when we used SpG at a six-fold higher concentration (8μM in the injection mix) we observed an efficiency of wrmScarlet targeting similar to that of SpCas9 (Fig. 1i).

Altogether, we demonstrate that SpG and SpRY, when using optimized delivery concentrations of mRNA or purified protein formulations, can exhibit high editing activities in distinct animal species grown at different temperatures. Furthermore, SpG and

**Fig. 1 SpG and SpRY are active nucleases in zebrafish and *C. elegans*. a** Two gRNAs (a, b) targeting *slc45a2* exon 1 in zebrafish (top). Experimental setup to analyze CRISPR-Cas9, SpG, and SpRY-mediated mutations in zebrafish by injecting one-cell-stage embryos (bottom). **b** Phenotypes obtained after the injection of the mRNA–gRNA duplex targeting *slc45a2* (albino) showing different levels of mosaicism (albino-like (alb-like), severe, mild) compared to the WT. Lateral views (scale bar, 1 mm) and insets of the eyes (scale bar, 0.2 mm) of 48 h post-fertilization (hpf) embryos are shown. **c** Percentage of albino-phenotypes (panel b) in embryos 48 hpf. (*n*) total number of injected embryos. The results were obtained from at least two independent experiments. **d** Phenotypic evaluation at different concentrations of gRNAs and mRNAs. Stacked bar plots show the percentage of the phenotypes described in panel b. The results were obtained from at least two independent experiments. **e** gRNAs were complexed with purified proteins to form RNPs for in vitro and in vivo testing in *C. elegans* by microinjection[65]. **f** Sequences of gRNAs targeting *dpy-10* with distinct complementarity. RNP combinations comprised of each gRNA and WT SpCas9, SpG, or SpRY were tested in vitro. Top bands show uncleaved PCR product. Lower bands show cleaved products. **g** The *dpy-10* matched and +5 gRNAs were tested for in vivo activity in *C. elegans* by injecting a single gonad arm. Each dot represents the editing efficiency in each $P_0$ that produced at least 100 $F_1$s. The results were obtained from two independent experiments, with both conditions carried out in parallel injections (Student's *t* test *p* value). **h** Schematic representation of in vivo experiments in *C. elegans* using a *gtbp-1::wrmScarlet* reporter by screening for loss of fluorescence. *dpy-10* gRNA was used for co-CRISPR. **i** An anti-wrmScarlet gRNA with NGG PAM was complexed with 1.3 μM of SpCas9, SpG, or SpRY to compare their in vivo efficiencies. In a separate experiment, the SpG-anti-wrmScarlet (NGG) RNP was injected at 8.0 μM. Each dot represents the editing efficiency in each $P_0$ that produced at least five Dpy or Rol $F_1$s. (One-way ANOVA followed by Tukey's test for multiple comparisons *p* values).

SpRY are equally sensitive to mismatches in the protospacer sequence as compared to SpCas9.

**Genome editing with SpG and SpRY across various zebrafish target sites**. To further evaluate SpG activity in vivo, we selected 15 targets with NGH (H = A, C, or T) PAMs in three genes whose loss-of-function (Fig. 2a) can be easily quantified by phenotype and correlates with the efficiency of different CRISPR-Cas systems[13,20]. As we previously optimized for targeting sites with NGG PAMs, we observed a strong enhancement of activity in most of the targets when increasing gRNA and mRNA con-centrations (Supplementary Fig. 5a–c). Notably, using our opti-mized conditions, we found high-efficiency activity in 6 out of 15 analyzed targets where at least 50% of the embryos presented severe (Class II or alb/gol severe mosaic) or extremely severe (Class III or alb/gol like) phenotypes, and 12 out of 15 targets showed at least 10% mosaic mutant embryos (Fig. 2b–d, Sup-plementary Data 1). These efficiency ratios were comparable to what was observed with WT SpCas9 in similar experiments[13]. In addition, by amplifying the genomic target regions from some of these mosaic mutant embryos, we identified DNA lesions induced by SpG activity that were similar to what is described for SpCas9 with short insertions or deletions (Supplementary Fig. 6a).

SpRY was previously shown to edit targets with NRN PAMs, but those with NGG and NGH PAMs can be edited with SpCas9 and SpG, respectively. Therefore, we focused our analysis of SpRY against eight genomic targets with NAN PAMs from the three loci that we previously used for SpG. Using our optimized conditions, we observed variable activity at the different targets where 3 out of 8 showed high activity (Fig. 2e, f, Supplementary Fig. 7a–c) and 6 out of 8 showed some activity. Notably, as we observed in *C. elegans*, phenotype quantification using SpG and SpRY correlated with genome mutagenesis in zebrafish where highly efficient targets showed the highest indel levels (Supple-mentary Fig. 7d and Supplementary Data 1). Indeed, highly efficient targets showed an average of 68.5% with a distribution from 40% to 91.5% (Supplementary Fig. 7d). Importantly, injections with SpCas9 mRNA showed very low penetrance or absence of phenotypes in the NGH or NAN targets (at least more than 90% WT embryos at the phenotype level) where SpG and SpRY performed efficiently, respectively (Supplementary Fig. 7e). This result reinforces the observations in mammalian cells where SpCas9 activity against targets with non-NGG PAMs is generally low or absent[6,21]. Finally, we tested whether temperature could affect SpG and SpRY activity in vivo. We did not observe any substantial difference in the activity of most of the gRNAs used (10 out of 11, Supplementary Fig. 7f) with SpG and SpRY at 28 °C and 34 °C, which is the highest temperature that allows normal

zebrafish development[20]. Altogether, SpG and SpRY can generate mutants at different temperatures in zebrafish embryos at genomic sites where SpCas9 is poorly active or inactive, with variable activity among different targets.

**Comparison of SpCas9, SpG, and SpRY across various targets in *C. elegans***. *C. elegans* is a model organism with convenient features for testing new Cas proteins in vivo[22]. We evaluated SpG and SpRY activities on NGH and NAN PAMs, respectively, for targeting *gtbp-1::wrmScarlet* (Fig. 3a). First, we checked whether RNP concentration is also critical for editing with SpG on a site with an NGH PAM. As has occurred at NGG PAM targets, SpG is more efficient when the concentration is increased from 1.3 to 3.7, and to 8.0 μM (Fig. 3b). Since increasing the volume of the nuclease in the injection mix is accompanied by a parallel increase in KCl concentration, we tested whether the latter has an effect on editing efficiency. In addition, some protocols deliber-ately increase the ionic strength of the injection mix to prevent RNP aggregation[23,24] whereas others do not[16,19,25]. By injecting six-fold higher (300 mM) KCl concentration in a CRISPR-SpG experiment targeting an NGH PAM while keeping the nuclease concentration constant, we observed that the higher salt con-centrations in the injection mix do not affect editing efficiency (Supplementary Fig. 8).

To compare the efficiency of SpCas9 and SpG for targeting sites with NGH PAMs, we performed experiments using these nucleases at the highest concentration (8 μM) to produce mutations in *gtbp-1::wrmScarlet*. We observed that SpG was more efficient than SpCas9 (mean of 62.1% vs 30.2%) when targeting sites with NGH PAMs (Fig. 3c). Interestingly, as previously reported in human cell lines[6], we found that SpRY is also active in targets with an NGH PAM in vivo, with similar efficiency to that of SpG (Fig. 3c). In contrast, SpG was previously reported to exhibit minimal activity against sites with NAN PAMs in human cells[6]. Therefore, we focused on the analysis of SpRY activity in three different NAN targets in the *gtbp-1::wrmScarlet* locus. First, we studied the in vitro capacity of SpCas9, SpG, and SpRY to cleave dsDNA targets with NAN PAMs. In vitro, SpRY had the highest cutting efficiency at all three target sites harboring an NAC PAM (Fig. 3d). As expected, SpG was also capable of cutting two of the targets in vitro. This may be due to the minor but still detectable SpG activity as previously observed at NAC sites in mammalian cells[6] (Fig. 3d). When tested in vivo, as we observed for SpG, higher concentra-tions of SpRY RNP significantly enhanced its activity in one of the analyzed targets (Fig. 3e). Finally, we compared the efficiency of SpRY and SpCas9 in one of the targets with a high level of

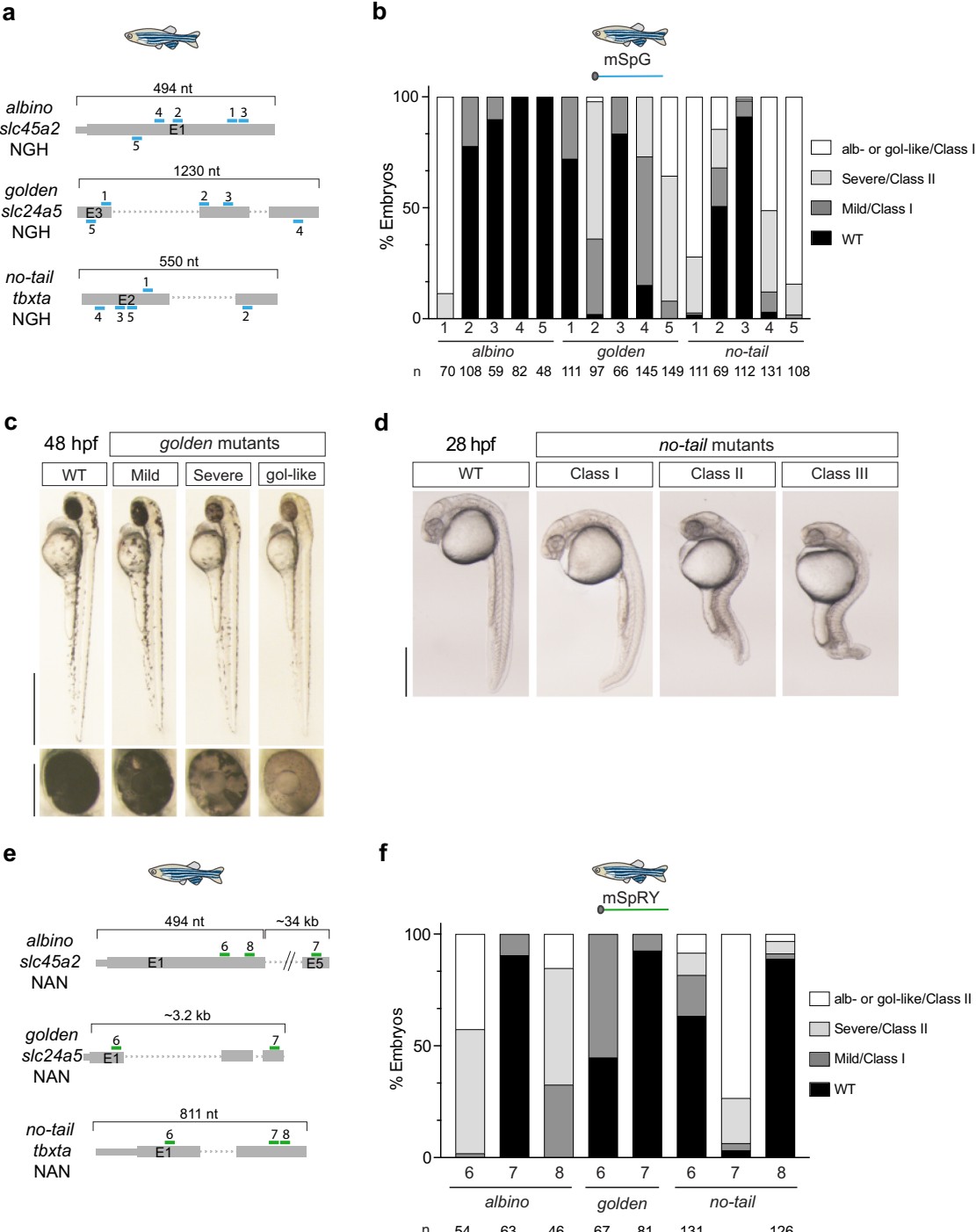

**Fig. 2 SpG and SpRY are active nucleases at minimal PAM targets in zebrafish. a** Diagram illustrating 15 gRNAs (blue) with NGH PAM in three zebrafish genes. Five gRNAs targeting exon 1, exons 3, 4 and 5 and exons 2 and 3 of *slc45a2, slc24a5* and *tbxta*, respectively. Different NGH PAMs for each target are detailed in Supplementary Data 1. **b** mRNA SpG shows high activity in NGH sites. Individual gRNAs described in panel a were injected with the SpG mRNA (240 pg gRNA and 300 pg mRNA per embryo). Stacked barplots show the percentage of albino-like/golden-like (gol-like)/class III, severe/class II, mild/class I, and phenotypically wild-type (WT) embryos 48 hpf after injection. (n) total number of injected embryos. The results were obtained from at least two independent experiments. **c** Phenotypes obtained after the injection of the mRNA–gRNA duplex targeting *slc24a5* showing different levels of mosaicism (golden-like, severe, mild) compared to the WT. Lateral views (scale bar, 1 mm) and insets of the eyes (scale bar, 0.2 mm) of 48 hpf embryos are shown. **d** Phenotypes obtained after injection of the mRNA–gRNA duplex targeting *tbxta* in zebrafish embryos (Lateral views). Levels of mosaicism compared to wild type (WT) were evaluated at 28 hpf. Class I: Short tail (least extreme). Class II: Absence of notochord and short tail (medium level). Class III: Absence of notochord and extremely short tail (most extreme). Scale bar, 0.5 mm. **e** Diagram illustrating 8 gRNAs (green) with NAN PAM in three zebrafish genes. Three gRNAs targeting *slc45a2* exons 1 and 5, two gRNAs targeting *slc24a5* exons 1 and 3 and three gRNAs targeting *tbxta* exons 1 and 2. Different NAN PAMs for each target are detailed in Supplementary Data 1. **f** mRNA SpRY shows high activity in NAN sites. Individual gRNAs described in panel e were injected with the SpRY mRNA (240 pg gRNA and 300 pg mRNA per embryo). Stacked barplots show the percentage of albino-like/golden-like/class III, severe/class II, mild/class I, and phenotypically wild-type (WT) embryos 48 hpf after injection. (n) total number of injected embryos. The results were obtained from at least two independent experiments.

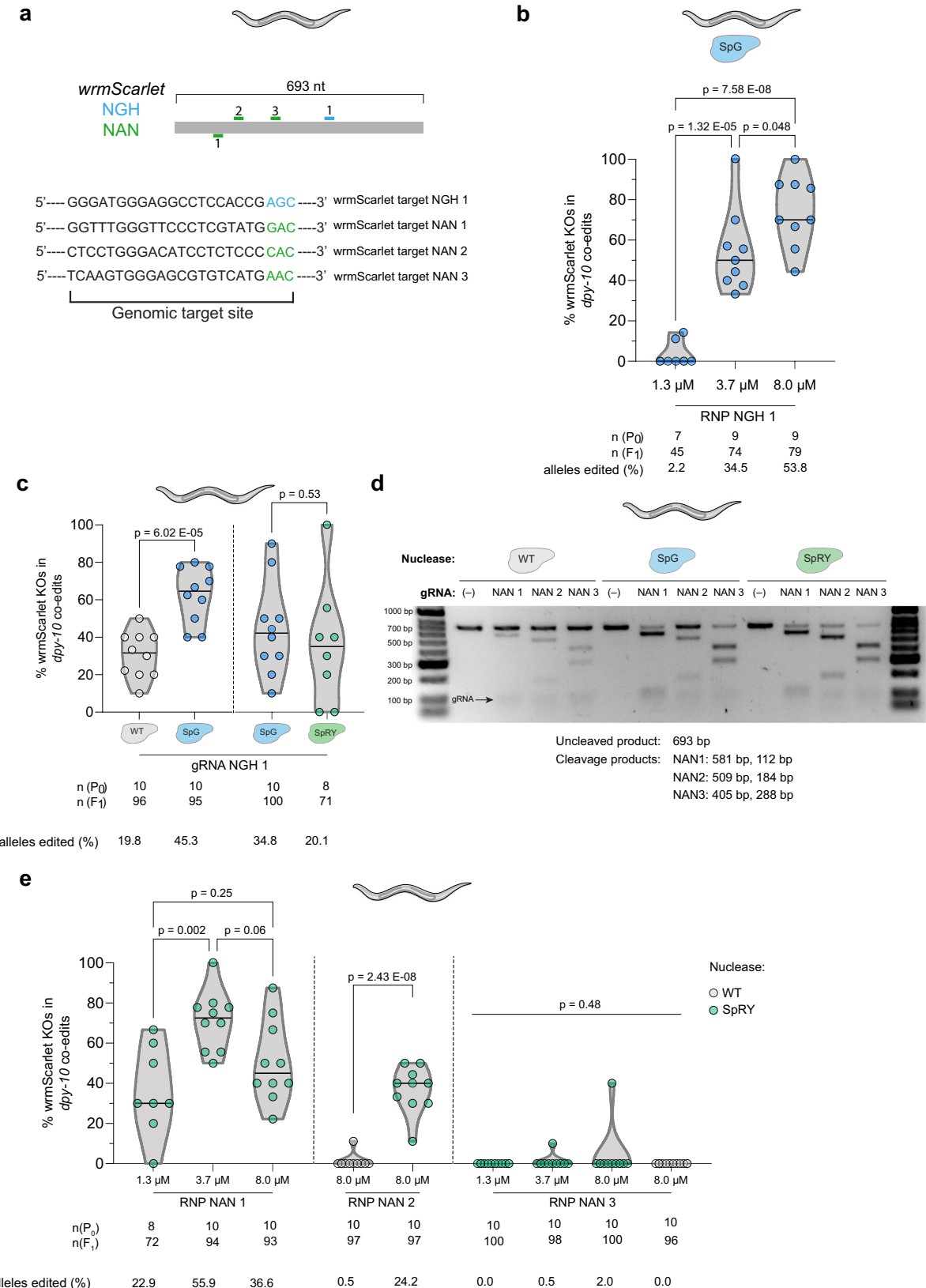

mutagenesis (NAN 2) and demonstrated that SpRY is much more effective than SpCas9 for editing NAN PAMs in vivo (Fig. 3e).

Together, these results are consistent with those obtained in zebrafish mRNA-injected embryos and suggest that increasing the concentration of SpG and SpRY RNPs enhances the activity of

these nucleases in *C. elegans*, particularly in NRN targets where SpCas9 has minimal activity. Furthermore, similar to our observation in zebrafish, the DSBs produced by SpG and SpRY lead to the formation of indels near the cut site, analogously to SpCas9 (Supplementary Fig. 6b), and each target site had variable

**Fig. 3 SpG and SpRY are active nucleases at minimal PAM targets C. elegans. a** Diagram illustrating one crRNA with NGH PAM (blue) and three crRNAs with NAN PAM (green) targeting wrmScarlet. The sequences of the crRNAs and PAMs are shown below. **b** Three distinct concentrations of SpG RNP, with the anti-wrmScarlet NGH gRNA 1, were tested in a strain expressing *gtbp-1::wrmScarlet*. Editing efficiency is defined as the number of $F_1$ worms exhibiting loss of fluorescence in the $F_2$ divided by the total number of separated *dpy-10* co-edited $F_1$s. Each dot represents the editing efficiency in each individual $P_0$ that produced at least ten Dpy or Rol $F_1$s. All three conditions were carried out in parallel injections (One-way ANOVA followed by Tukey's test for multiple comparisons *p* values). **c** Comparison of editing efficiencies with anti-wrmScarlet NGH gRNA 1 between SpCas9 and SpG, and SpG and SpRY in independent experiments at an RNP concentration of 8.0 µM. Editing efficiency, dots, and numbers are defined as in panel b. Conditions belonging to parallel injections are separated by a dashed line (Student's *t* test *p* value). **d** In vitro analysis of three anti-wrmScarlet NAN gRNAs. Different RNP combinations were tested in vitro at 37 °C by incubating the RNPs with wrmScarlet PCR product. Top row of bands shows uncleaved PCR products and the specific cleavage products for each gRNA are specified in the figure. The gRNA appears as a faint band at approximately 100 bp. This experiment was performed once. **e** In vivo analysis of the three anti-wrmScarlet NAN gRNAs. A titration of three distinct SpRY RNP concentrations was performed for gRNAs 1 and 3, while gRNA 2 was tested at 8.0 µM only. The editing efficiency of SpCas9 in NAN sites was evaluated using gRNAs 2 and 3. Editing efficiency, dots, and numbers are defined as in panel b. A dashed line separates conditions belonging to parallel injections (One-way ANOVA followed by Tukey's test for multiple comparisons [NAN 1 and 3], Student's *t* test [NAN 2] *p* values). The percentage of alleles edited in panels b, c, and e was calculated as previously described in Fig. 1i.

activity that likely depends on the genomic target site and nucleotide context adjacent to the PAM as described for Cas9 in different models[6,13,26,27].

**CRISPRscan can help to predict SpG and SpRY highly efficient targets.** As previously described for wild-type SpCas9[13,26,28], we observed variability among the activities at different sites targeted with SpG or SpRY. Algorithms predicting CRISPR-Cas9 activity have strikingly contributed to the selection of highly efficient gRNAs that significantly increase the effectiveness of editing[13,26,28]. CRISPRscan is a convenient tool for predicting CRISPR-SpCas9 activity in vivo that has been particularly useful when gRNAs are transcribed in vitro or chemically synthesized, and are co-delivered with Cas9 mRNA[13,28]. This is a commonly used approach, not only in zebrafish injections, but also in other vertebrates such as *Xenopus* or mouse[13,28]. Since SpG and SpRY are modified versions of SpCas9 and no clear biases have been observed in terms of PAM preference for these Cas enzymes[6,29–31], we tested whether the CRISPRscan algorithm built for SpCas9 would help to predict the activity of these minimal PAM nucleases in zebrafish. We calculated the CRISPRscan score of the 17 and 10 targets used for SpG (NGN) and SpRY (NGG and NAN), respectively. Targets with a CRISPRscan score of 66 or more showed significant enrichment of highly efficient mutagenesis with at least 50% of the embryos demonstrating a severe (Class II/severe mosaic) or extremely severe (Class III/ albino or golden-like) phenotype (Fig. 4a). These results suggest that CRISPRscan can help to select the most active gRNAs for SpG and SpRY in vivo when using in vitro transcribed gRNAs. This can significantly increase the number of highly efficient targets not only in open reading frames but also in small genomic regulatory sequences such as those coding mature miRNAs (Supplementary Fig. 9).

The lower number of tested targets and the different gRNA formulations in *C. elegans* (crRNA:tracrRNA vs gRNA in zebrafish) precluded us from running a similar analysis but we found a correlation between efficiency and CRISPRscan scoring in the wrmScarlet locus, in which we tested five targets. For example, while the NAN3 target with a CRISPRscan score of 39 (Supplementary Data 1) showed very low activity even in optimized conditions (4%), targets predicted to be highly efficient (NAN 1, NAN 2, NGH1 (Fig. 3b, e) and NGG (Fig. 1h)), with scores of more than 70 (Supplementary Data 1), showed at least a nine-fold increase in activity. In any case, a different approach was used in *C. elegans* where RNPs were injected, which can also influence the final SpG or SpRY activity since RNPs can protect from in vivo gRNA degradation and increase the half-life of unstable gRNAs[20,32,33]. Indeed, by using SpG or SpRY RNPs in zebrafish we were able to recapitulate the results of mRNA–gRNA

injections and to increase the penetrance of the phenotypes in some targets where we previously observed low activity (Fig. 4b, Supplementary Fig. 10). Altogether, these results show that CRISPRscan predictions for SpG and SpRY highly efficient targets could facilitate the use of these nucleases in zebrafish and demonstrate that RNP formulations can enhance editing activity particularly when poorly stable gRNAs are used.

**SpG and SpRY are suitable for HDR-mediated gene editing in *C. elegans*.** To further support the use of SpG and SpRY in animals, we studied the efficiency of SpG and SpRY for precise HDR-mediated genome editing by generating missense mutations and fluorescent reporters in *C. elegans*[34]. First, by choosing an NGN PAM target closer to the edit of interest compared to a site with an NGG PAM, we introduced the substitution R350C (which mimics a human cancer mutation) in the *C. elegans* protein SWSN-4/SMARCA4 with an efficiency of 10% among *dpy-10* co-edited animals (Fig. 4c and Supplementary Data 2). Then, also using SpG, we inserted a partial wrmScarlet sequence at the C-terminal end of two genes (*usp-48* and *trx-1*) lacking an NGG PAM proximal to the stop codon (Fig. 4c). This corresponds to the Nested CRISPR strategy[19], where the insertion of a short sequence (≤200 bp ssDNA as repair template; step 1) facilitates the insertion of a longer fragment, using dsDNA as a repair template, to complete the fluorescent protein sequence (step 2). Thus, in the first step of Nested CRISPR, among *dpy-10* co-edited worms, we obtained 19.7% and 13.1% of inserts at the *usp-48* and *trx-1* C-terminal ends, respectively. We also generated a GFP::H2B transcriptional reporter by replacing the entire W05H9.1 coding sequence for the GFP::H2B tag. In this case, we used SpCas9 to cut at the C-terminal end, and SpG to cut at the N-terminal end (Fig. 4c). The step 1 Nested CRISPR efficiency for this reporter was 6.6% among *dpy-10* co-edited worms. Also using ssDNA as donor template, we made a translational reporter for *cep-1*, using SpRY and the stop codon TAA as NAN PAM (efficiency of 7.1% among co-edited worms), and tagged *cki-1* with a degron sequence[35] with a 200-bp ssDNA repair template (efficiency of 16.6% among co-edited worms) (Fig. 4c). Altogether, we demonstrate that SpG and SpRY can be used for HDR insertions at NGN and NAN sites, respectively.

Finally, we used a 752 bp dsDNA fragment as a donor template to generate GFP reporters at the *gtbp-1* locus with SpG and SpRY through the second step of Nested CRISPR at an NGG PAM[19]. An editing efficiency of 21.7% was achieved using a commercial Cas9 (IDT) at 1.58 µM[19], similar to the 22.9% obtained by using SpG at 3.8 µM (Fig. 4c).

The widespread use of Cas9 variants in *C. elegans* can be facilitated by providing transgenic strains that endogenously

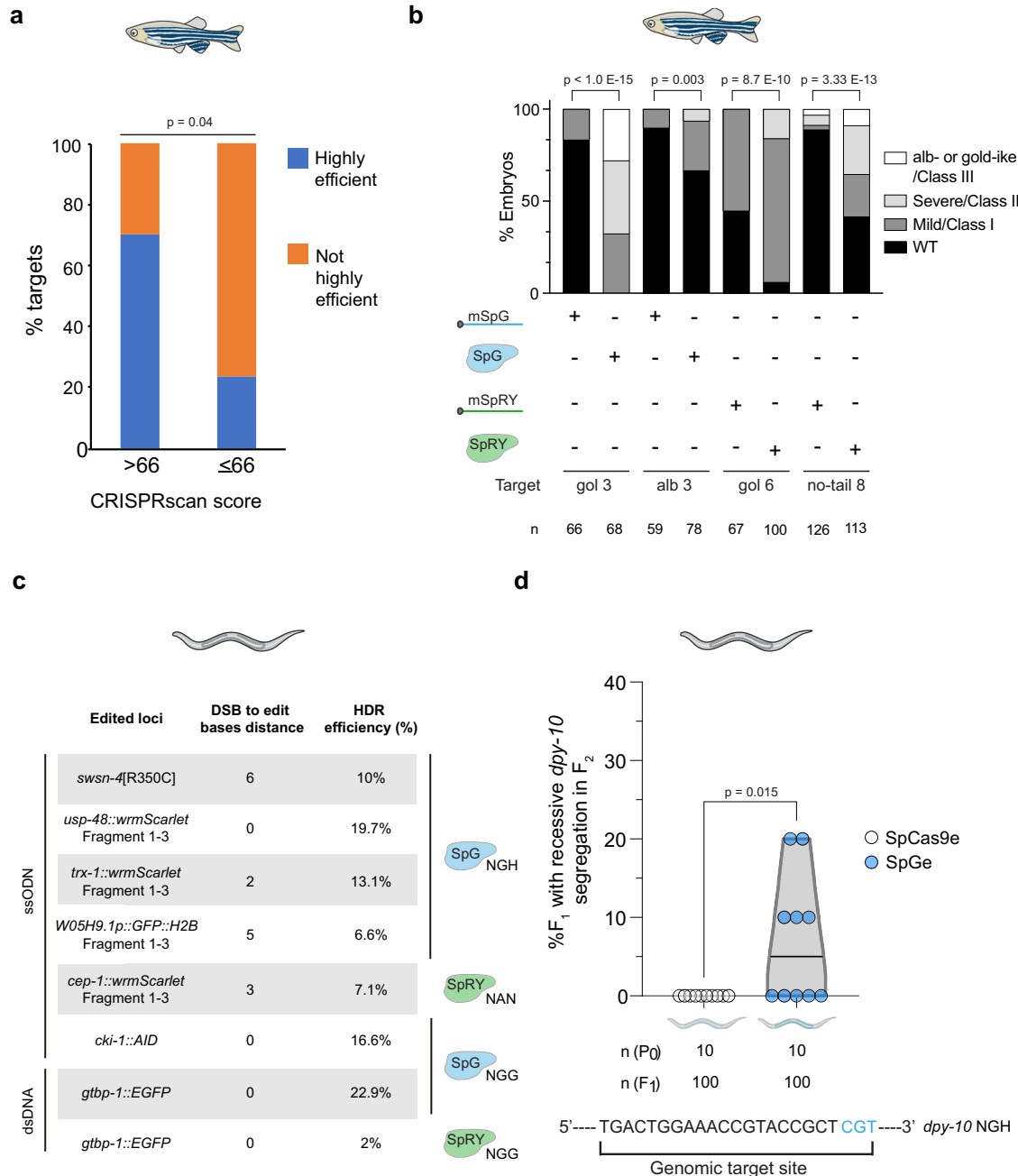

**Fig. 4 Further SpG and SpRY optimization in vivo. a** Prediction of gRNA activity in vivo using CRISPRscan. Stacked barplots show the percentage of highly efficient gRNAs (blue) and not highly efficient gRNAs (orange) in two groups separated based on CRISPRscan scores (>66 and ≤66). Highly efficient gRNAs generate more than 50% of embryos with albino-like/golden-like/class III or severe/class II phenotypes and less than 10% phenotypically wild-type. Fisher test *p* value. **b** RNP can enhance SpG and SpRY activity in zebrafish. Stacked barplots show the percentage of albino-like/golden-like/class III, severe/class II, mild/class I, and phenotypically wild-type (WT) embryos 48 h post-fertilization (hpf) after injection. (*n*) total number of injected embryos. The results were obtained from at least two independent experiments. mSpG and mSpRY injections data from Fig. 2b, f, and RNP injections data from Supplementary Fig. 10. The χ2-test *p* value per comparison is shown. **c** Utility of SpG and SpRY for the insertion of DNA sequences via HDR. We used ssDNA (ssODN) as a donor template to produce a missense mutation in *swsn-4*, to introduce a gene fragment for nested CRISPR[19] at *usp-48*, *trx-1*, W05H9.1, and to add a degron tag to *cki-1*. Information about targeted sequences, PAMs, DSB to edit distance, insert lengths, nuclease concentration used, and correct vs incorrect insertions obtained, is showed in Supplementary Data 2. **d** EG9615 and CER660 worms, which express SpCas9 and SpG endogenously in the germline (SpCas9e and SpGe), respectively, were injected with tracrRNA and a crRNA targeting *dpy-10* with an NGH PAM. The fluorescent markers *myo-2p::mCherry* and *myo-3p::mCherry* were used as co-injection markers. F₁ worms expressing mCherry in the pharynx or body wall muscle were singled out and the appearance of Dpy progeny was screened in the F₂. Editing efficiency is defined as the number of F₁ worms that segregate Dpy progeny in the F₂ divided by the total number of separated F₁s. Each dot represents the editing efficiency in each individual P₀ that produced at least ten mCherry-expressing F₁s. All conditions were carried out in parallel injections (Student's *t* test *p* value).

express these nucleases in the germline. Starting from a strain expressing SpCas9 in the germline (SpCas9e)[36,37], we introduced mutations by CRISPR to obtain a modified strain that produces SpG instead of SpCas9. By injecting just crRNA and tracrRNA, we tested the mutagenic capacity of these two strains on an NGH *dpy-10* PAM target and observed that animals expressing SpG endogenously in the germline (SpGe) were more efficient than those expressing SpCas9 (Fig. 4d).

In summary, we have demonstrated that the near-PAMless Cas9 variants SpG and SpRY can mediate precise genome editing by HDR and have generated a strain expressing SpG endogenously to facilitate genome editing in targets with NGN PAMs.

## Discussion

Despite the tremendous impact of CRISPR-Cas9 in biotechnology and biomedicine, PAM requirements limit the technology for certain purposes. PAM proximity to the cut site is necessary when targeting short sequences such as microRNA or editing at specific sites by homologous recombination, or base or prime editing[38]. Such proximity is also required when CRISPR-Cas is used to diagnose specific mutations[39]. Investigations into bypassing these restrictions imposed by PAMs are centered on the search for Cas9 orthologs in nature and on engineering Cas9 variants[40]. Among engineered Cas9 variants, the near-PAMless nucleases, SpG and especially SpRY variants, present the most relaxed PAM requirements to date. The activity of these nucleases has been well described in human cell lines[6], and more recently in plants[30,31,41,42], Dyctiostelium[43], and *Candida albicans*[44]. However, SpG and SpRY have never been applied to animals. Here, we have demonstrated that both SpG and SpRY are competent in a vertebrate and an invertebrate animal model, zebrafish and *C. elegans*, respectively. Furthermore, we have shown that these near-PAMless nucleases are capable of generating mutations in cells in two different developmental contexts such as embryonic cells and germ cells in zebrafish and *C. elegans*, respectively. In addition, we have optimized two technical approaches not previously tested with these enzymes such as the use of purified mRNA or protein. This can be useful to avoid potential issues with plasmid integrations in other biological contexts as it has been recently observed using SpG and especially SpRY in mammalian cell cultures[29].

Notably, we observed that the relaxed PAM requirements of SpG and SpRY come with a cost in terms of in vivo editing efficiency. When using similar standard concentrations of CRISPR reagents (Cas9 and gRNA), SpG and SpRY did not perform as successfully as SpCas9. However, we showed that SpG and SpRY activity benefits from an increased concentration of CRISPR-Cas reagents. Such need could be due to the more extensive target scanning that occurs when more PAMs fit with their requirements[45]. We speculate that this reduced SpG and SpRY activity was not detected in human cell cultures or plant experiments because the gRNA and nuclease were expressed from strong, constitutive promoters that likely saturated their concentrations within the cells. Thus, our results in these two animal models indicate that SpG and SpRY exhibit a trade-off between versatility and efficiency that could be balanced by increasing the concentration of the gRNA, mRNA, and/or protein.

A possible drawback when using SpG and SpRY is the potential increase in off-targeting[6]. Importantly, we have observed that WT SpCas9, SpG, and SpRY are equally sensitive to mismatches at the protospacer sequence, and therefore such off-targeting could be predicted by current algorithms[26]. Moreover, our optimized transient approaches, such as RNP or mRNA–gRNA delivery, will help reduce potential off-targets compared to when gRNA and Cas9 are expressed from plasmids or stable integrations in the genome that provide a longer window of opportunity for off-

target mutagenesis. Finally, high-fidelity versions can help to decrease off-target activity in vivo as shown in mammalian cells where an increase in the specificity with only a slight reduction in the on-target activity was observed[6].

Since SpG and SpRY are engineered SpCas9 variants, we speculate that most of the optimizations uncovered and used for SpCas9 will be likely applicable to SpG and SpRY. Indeed, we have shown that CRISPRscan, an algorithm for predicting CRISPR-SpCas9 activity in vivo, would facilitate the selection of highly efficient targets for SpG and SpRY. CRISPRscan scores above 70 normally identifies highly efficient targets for WT SpCas9[13] similarly to our observations for SpG/SpRY where targets with scores of 67 or above showed high activity in a significant manner. Since our study is limited to 40 sites (25 in zebrafish and 15 in *C. elegans*), more studies will be needed to further evaluate predictive tools for SpG and SpRY activities. Meanwhile, we have updated our web tool (www.crisprscan.org) with the targets and predicted on- and off-target scores for SpG (NGN) and SpRY (NRN). Moreover, chemically modified and more stable gRNAs[46] can contribute to an improvement in editing in vivo, especially when co-injected with mRNA where gRNAs spend more time unprotected from degradation. Finally, Cas9 RNPs with synthetic crRNA:tracrRNAs used here for *C. elegans* could also increase the mutagenesis level in zebrafish[47].

Furthermore, recent publications have demonstrated not only the capacity of SpG and SpRY to produce indels in mammalian cells and plants but also their adaptation to base editor systems that take advantage of their relaxed PAMs[6,31,41]. Our optimized CRISPR-SpG and SpRY systems in vivo now set the basis for carrying out base editing and other CRISPR applications in animals[48–50] with a greatly improved targeting landscape. Indeed, we went one step further in using SpG and SpRY in *C. elegans* by performing editing through HDR that allows the mimicking of missense mutations and the creation of endogenous reporters in targets that are inaccessible for SpCas9. Finally, and as proof of principle, we have also produced an SpG transgenic nematode that will facilitate the use of the technology in this animal. This strategy, combined with the use of tissue-specific promoters and inducible gene expression systems, opens the possibility of taking advantage of these nucleases in the cell types of interest and at distinct developmental stages. Altogether, we believe that our SpG and SpRY optimizations will contribute to the expansion of the CRISPR-Cas toolbox for in vivo applications.

## Methods

**Number of target sites calculation for different Cas nucleases in zebrafish and *C. elegans* genomes.** Genome sequences from *C. elegans* (v. WBcel235) and *D. rerio* (v. GRCz11) were downloaded from Ensembl database[51].

PAM sites (Supplementary Data 1) were searched by an in-house script written in PERL language. It is based on the fuzznuc function of EMBOSS package to show the coordinates of the sites in both strands[52]. Genomic positions were converted into GTF format for later comparison to the different types of regions of the genome using bedtools[53]. The count of sites by genomic position was depicted by the ggplot2 3.3.0 library from R programming language.

To calculate the average of targets or CRISPRscan score >66 targets per CDS (Supplementary Fig. 9), gene annotations were imported from Ensembl release 104 using FONtools [https://github.com/vejnar/fontools]. Custom Python scripts were reading genome sequence using pyfaidx [https://github.com/mdshw5/pyfaidx], searching gRNA targets with regular expressions, and mapping to gene annotations from FON1 [https://github.com/vejnar/fontools]. CRISPRscan scores were obtained from [https://crisprscan.org]. Violin plots were plotted using Matplotlib [https://matplotlib.org].

**Target and gRNA design in zebrafish.** Target (protospacers) and gRNAs for SpG and SpRY were designed using an updated version of the algorithm CRISPRscan (www.crisprscan.org) tool[13]. Regular expressions for SpG (NGN) and SpRY (N[AG]N) were added for on-target and off-target searches. On-target scores were evaluated using the same CRISPRscan scoring algorithm used for SpCas9 using NGG as the PAM sequence. Off-targets with mismatches restricted, or not, to the target "seed" are reported similarly, to SpCas9[54,55]. The CFD score[26] used to

characterize potential off-targeting was kept identical to score protospacers but adapted to score PAMs of potential off-targets. SpG and SpRY matching PAMs were set a 1.0 score instead of using the original (lower than 1.0) Doench et al. score characterized for SpCas9. Protospacers in the different loci were selected without predicted off-targets[54], and within functional domains of the protein and in exons in the first half of the ORF with the exception of gRNA albino 7 which approximately maps in the last third part of the ORF. To evaluate activity of the Cas9 variants in the NGG PAMs, we used two gRNAs (albino a and albino b) as previously employed to evaluate SpCas9 activity[13]. All the information about the targets is detailed in Supplementary Data 1.

**In vitro transcription and gRNA generation.** gRNAs were generated as previously described[56]. gRNA DNA templates were amplified by fill-in PCR. Briefly, a 52-nt oligo (sgRNA primer), containing the T7 promoter, the 20 nt of the specific gRNA DNA binding sequence (spacer) starting with two Gs and a constant 15-nt tail for annealing, was used in combination with an 80-nt reverse oligo to add the gRNA invariable 3′ end (universal primer). Here, all spacers used in zebrafish experiments started with 5′ GG (Supplementary Fig. 1a) allowing 100% match between target and in vitro transcribed gRNAs. A 117 bp PCR product was generated following these parameters: 30 s at 98 °C, 30 cycles of 10 s at 98 °C, 30 s at 51 °C, and 30 s at 72 °C, and a final step at 72 °C for 1 min. PCR products were purified using FavorPrep™ GEL/PCR Purification kit (Favorgen) columns and approximately 120–150 ng of DNA was used as template for a T7 in vitro transcription (IVT) reaction (AmpliScribe-T7-Flash transcription kit from Epicentre). In vitro transcribed gRNAs were DNAse-treated using TURBO-DNAse for 20 min at 37 °C and precipitated with sodium acetate/ethanol and resuspended in RNAse and DNAse free water. gRNAs were visualized in 2% agarose stained with ethidium bromide to check for RNA integrity, quantified using the Qubit RNA BR Assay Kit (ThermoFisher, Q10210), and stored in aliquots at −80 °C.

**Target and gRNA design in C. elegans.** For experiments in the *dpy-10* locus, the *dpy-10* gRNA with NGG PAM used in co-CRISPR was used as the reference sequence[18]. Mismatches were then introduced into the protospacer at one (+1) or five (+5) nucleotides upstream of the PAM (Fig. 1f). Meanwhile, a protospacer with an NGH PAM was chosen based on proximity to the reference NGG sequence to maintain the cut site within the RXXR domain of *dpy-10* which is responsible for the production of the dominant dumpy and roller phenotypes, such as that of the *cn64* allele[57]. Five targets with distinct PAM requirements were selected for wrmScarlet: one NGG, one NGH, and three NAN (Fig. 3a). For HDR experiments in *swsn-4*, *usp-48*, *trx-1*, W05H9.1, *cep-1*, and *cki-1*, gRNAs were selected based on the proximity of the DSB from the desired edit site (Fig. 4d). To generate the *gtbp-1*::EGFP reporter with SpG, the Nested CRISPR universal EGFP step 2 crRNA was used[19]. crRNAs and ssODNs were purchased in tubes from IDT as 2 nmol ALT-R crRNAs and 4 nmol ultramers, respectively, and resuspended in 20 μl or 40 μl of nuclease-free duplex buffer (30 mM HEPES, pH 7.5; 100 mM potassium acetate. IDT, Cat. No. 11-01-03-01), respectively, to yield a stock concentration of 100 μM. A list of all crRNAs used can be found in Supplementary Data 1.

**SpG and SpRY zebrafish codon optimized constructs and mRNA generation.** pT3TS_zCas9 (Addgene, 46757)[58] was modified to generate two plasmids encoding the SpG and SpRY zebrafish codon-optimized variants: pT3TS_zSpG and pT3TS_zSpRY include D1135L/S1136W/G1218K/E1219Q/ R1335Q/T1337R and A61R/L1111R/D1135L/S1136W/G1218K/E1219Q/ N1317R/A1322R/R1333P/ R1335Q/T1337R modifications, respectively. All mutations were generated using site-directed mutagenesis (QuikChange Multi Site-Directed Mutagenesis Kit, Agilent Technologies) and primers used are detailed in Supplementary Data 1. The final pT3TS_zSpG (Addgene #179316) and pT3TS_zSpRY (Addgene #179317) constructs were confirmed by sequencing.

SpG and SpRY mRNA were in vitro transcribed from DNA linearized by XbaI (1 μg) using the mMESSAGE mMACHINE™ T3 kit (Invitrogen, Thermo Fisher). In vitro transcribed mRNAs were DNAse treated using 1 μL TURBO-DNAse for 20 min at 37 °C and purified using RNeasy Mini Kit (Qiagen). mRNA product was quantified using NanoDrop (Thermo Fisher) and stored in aliquots at −80 °C.

**SpG and SpRY purification.** The two Cas9 variants: SpG (D1135L/S1136W/ G1218K/E1219Q/R1335Q/ T1337R) and SpRY (A61R/L1111R/D1135L/S1136W/ G1218K/E1219Q/ N1317R/A1322R/R1333P/R1335Q/T1337R) were cloned into the pET-28b- Cas9-His, with DNA gBlocks (IDT) encoding for the mutated regions (Addgene #179318 and #179320).

The different Cas9 proteins were expressed in *E. coli* (DE3) using the auto-induction method, by growing for 4 h at 37 °C, followed by 20 h expression at 25 °C. Cells were harvested by centrifugation at 4000 × g for 10 min. Cell pellets were suspended in equilibration buffer (20 mM Tris pH 8.0, 500 mM NaCl) plus protease inhibitors (Roche) and lysed using a high pressure Emulsiflex. The cell debris were removed by centrifugation at 30000 × g for 30 min. The supernatant was purified using a HisTrap column (Cytiva) pre-equilibrated with buffer A (20 mM Tris pH 8.0, 500 mM NaCl). After sample loading, the columns were washed with buffer A plus 50 mM imidazole and eluted with buffer A plus 500 mM imidazole. The eluted protein was concentrated with Amicon Ultra 50 Kda filters

(Millipore) and loaded into a Superdex 200 10/300 size-exclusion column (Cytiva) equilibrated with SEC buffer (20 mM HEPES pH 7.4, 500 mM KCl, and 1 mM DTT). The eluted sample was dialyzed against storage buffer 20 mM Tris pH 7.4, 200 mM KCl, 10 mM MgCl₂, and 10% glycerol, concentrated to 2 mg/mL, snap frozen in liquid nitrogen, and stored at −80 °C.

**RNP in vitro test.** The gRNA was prepared by pre-annealing 3.2 μL of 32 μM ALT-R tracrRNA (IDT, Cat. No. 1072532) and 1 μL of 100 μM crRNA with 5.8 μL of nuclease-free duplex buffer (IDT) at 95 °C for 5 min. Dilutions of the components, namely gRNA, nuclease, and PCR product were prepared at 300 nM, 900 nM, and 90 nM, respectively. Then, the RNP complex was assembled by incubating 9 μL of gRNA with 3 μL of nuclease in 12 μL of nuclease-free H₂O with 3 μL of 10x Cas9 reaction buffer (New England Biolabs, #B0386) at 37 °C for 15 min. Alt-R® S.p. Cas9 Nuclease V3 (IDT, Cat. No. 1081058) was used as commercial Cas9. 3 μL of the DNA substrate (PCR product) containing the target site was then added to achieve a final molar ratio of nuclease, gRNA, and target site of 10:10:1 (90 nM:90 nM:9 nM). The 30-μl reactions were incubated at different temperatures (15, 25, 37, and 50 °C) for 60 min. To release the DNA substrate from the RNP complex, 1 μl Proteinase K (20 mg/mL) was added to the reaction and incubated at 56 °C for 10 min. The cleaved products were analyzed through agarose gel electrophoresis using a 2% gel stained with SYBR® safe DNA gel stain (ThermoFisher Scientific, Cat. No. S33102).

**Zebrafish maintenance.** Wild-type zebrafish embryos were obtained through natural mating of AB/Tübingen AB/Tu zebrafish of mixed ages (5–18 months). The selection of mating pairs was random from a pool of 20 males and 20 females. All experiments involving zebrafish conform to national and European Community standards for the use of animals in experimentation and were approved by the ethical committees from the University Pablo de Olavide, CSIC, and the Andalusian Government. Zebrafish wild-type strains AB/Tübingen (AB/Tu) were maintained and bred under standard conditions[59]. All experiments were carried out at 28 °C and 34 °C, temperatures allowing optimal zebrafish development.

**C. elegans maintenance.** We used the Bristol N2 strain as the wild-type background while the strain CER541 *gtbp-1(cer149[gtbp-1::wrmScarlet]) IV* was used for wrmScarlet knockout experiments. Worms were maintained at 15 °C, 20 °C, or 25 °C on Nematode Growth Medium (NGM) plates seeded with *Escherichia coli* OP50 bacteria[60]. All strains generated in this study are listed in Supplementary Data 1.

**mRNA and RNP injections and image acquisition in zebrafish.** mRNA injection mixes were prepared at two different concentrations by combining the mRNA of the variants, mSpG and mSpRY, and the gRNAs. One nL containing 150 pg of SpG or SpRY mRNA and 20 pg of gRNA were injected into one-cell stage embryos, similar to what has been previously used for SpCas9[13]. To optimize SpG and SpRY activity, either 0.5 nL or 1 nL of a mixture containing 300 pg of mRNA and 240 pg of gRNA or 1 nL of a mixture containing 600 pg of mRNA and 480 pg of gRNA was injected. RNP injection mixes were prepared in 20 mM HEPES pH 7.4, 250 mM KCl, and 1 mM DTT by mixing the protein and gRNA at a ratio of 1:1.3. RNPs (6 μM) were incubated at 37 °C for 10 min and then kept on ice before use. One nL (6 fmol), or 0.5 nL (3 fmol) from the 6 μM solution was injected in one-cell stage embryos. In both cases, the mixtures were kept on ice, and any excess stored at −80 °C for up to three freeze-thaw cycles maintained similar efficiency. Zebrafish embryos were incubated at 28 °C and phenotypes were analyzed at 28 h or at 48 h post-fertilization, depending on the target gene, using an Olympus SZX16 stereoscope and photographed with a Nikon DS-F13 digital camera and further edited in Adobe Photoshop. When analyzing the temperature effect in SpG/ SpRY targeting, zebrafish embryos were incubated at 34 °C for 24 hpf and then changed to 28 °C until the end of the experiment. Both 34 °C and 28 °C temperatures allow normal development[20].

**Microinjection in C. elegans.** Injection mixes were prepared by combining equimolar concentrations of Cas9 nuclease, tracrRNA, and crRNA; and incubated at 37 °C for 15 min. When necessary, ssODN repair templates were added after incubation and the mixture centrifuged at 16,100 × g for 2 min to settle particulate matter. The injection mixes were kept on ice prior to loading the needles and any excess stored at −20 °C afterwards. Eppendorf Femtotips® capillary tips (Eppendorf, Cat. No. 930000035) for microinjection were loaded with 2 μL of the injection mix and fixed onto the *XenoWorks* Microinjection System (Sutter Instrument) coupled to a Nikon Eclipse Ti-S inverted microscope with *Nomarski* optics. Approximately 15–20 young adult hermaphrodites were injected for each experimental condition. The worms were fixed on 2% agarose pads with halocarbon oil in groups of five and were injected in one or both gonad arms. Injected worms were recovered in M9 buffer and were individually separated onto nematode growth medium (NGM) agar plates. The plates were incubated at 25 °C for 3 days.

**RNP testing in C. elegans.** Bristol N2 worms were injected with *dpy-10* RNPs and screened for the presence of dumpy and/or roller phenotypes. The editing

efficiency from each injected $P_0$ was calculated by counting the proportion of dumpy or roller $F_1$ progeny over the total number of $F_1$ progeny laid by each $P_0$ worm. Occasionally, injections targeting the *dpy-10* locus included pCFJ90 (*myo-2p*:: mCherry) and pCFJ104 (*myo-3p::mCherry*) as co-markers to facilitate the screening of recessive Dpy phenotypes in the $F_2$. On the other hand, CER541 worms harboring a homozygous *gtbp-1::wrmScarlet* fluorescent reporter were injected with anti-wrmScarlet RNPs combined with *dpy-10* RNP as co-CRISPR marker. From each injected $P_0$, between five to ten dumpy or roller $F_1$s were separated and allowed to lay $F_2$ progeny. The $F_2$ progeny were then screened for wrmScarlet knockouts using a Nikon SMZ800 stereomicroscope linked to a Nikon Intensilight C-HGFI epi-fluorescence illuminator with an mCherry filter. The editing efficiency from each injected $P_0$ was calculated by counting the proportion of separated $F_1$ progeny that gave rise to non-fluorescent $F_2$ worms. Non-fluorescent worms were indicative of indels arising from error-prone repair of DSBs. The percentage of edited alleles was calculated as (2 × no. of homozygous F1 knockouts + no. of heterozygous F1 knockouts)/2 × total no. of F1 screened × 100.

Endogenous reporters for the *usp-48*, *trx-1*, and W05H9.1 loci were generated to test the efficiency of HDR by using SpG with NGN PAMs, and in the *cep-1* locus by using SpRY with an NAN PAM. Using the Nested CRISPR approach[19], a C-terminal wrmScarlet fusion was made in *usp-48*, *trx-1*, and *cep-1*, while a transcriptional reporter was made for W05H9.1 by removing the entire coding sequence and replacing it with the GFP::H2B sequence. For each locus, an injection mix containing the SpG nuclease, target gene crRNA(s), *dpy-10* crRNA, tracrRNA, and an ssODN repair template consisting of a partial wrmScarlet or GFP::H2B fragment flanked by two 35-bp homology arms were assembled and injected into Bristol N2 worms. *dpy-10* co-edited $F_1$ progeny were then separated individually or in pools onto NGM agar plates, allowed to lay $F_2$ progeny, and genotyped via PCR. The overall editing efficiency was calculated by counting the proportion of $F_1$ worms harboring the insertion of the correct size over the total number of genotyped $F_1$ animals. At least two independent homozygous lines for the step 1 insertion are then verified via Sanger sequencing and the complete wrmScarlet or GFP::H2B fragment is inserted via the nested CRISPR step 2 protocol using SpCas9[19]. On the other hand, the R350C substitution in the *swsn-4*/SMARCA4 gene and the tagging of *cki-1* at the C-terminus with a degron sequence was accomplished using SpG and ssODN repair templates. Finally, the *gtbp-1*::EGFP reporter was generated with SpG following the Nested CRISPR step 2 protocol[19], but with a nuclease concentration for 3.8 μM instead of 1.6 μM. A list of primers used for genotyping and sequencing can be found in Supplementary Data 1.

**Generation of endogenous germline SpG-expressing *C. elegans* strains**. EG9615 (*oxSi1091[mex-5p::Cas9(smu-2 introns) unc-119(+)] II; unc-119(ed3)* III), a strain carrying a transgene that expresses SpCas9 in the germline was a gift from Dr. Matthew Schwartz and Dr. Erik Jorgensen[37]. EG9615 hermaphrodites were injected with three crRNAs and three ssODN repair templates to introduce the six amino acid substitutions to convert SpCas9 to SpG. Each crRNA and ssODN repair template introduced the D1135L and S1136W, G1218K and E1219Q, and R1335Q and T1337R substitutions by pairs (Supplementary Data 1). The first round of injections contained all three crRNAs at a final concentration of 1 μM each, tracrRNA at 3.2 μM, the three ssODN repair templates at 2.2 μM each, pCFJ90 at 2.5 ng/μL, and pCFJ104 at 5.0 ng/μL. $F_1$ progeny with visible mCherry expression in the pharynx or body wall were singled out and were genotyped via single worm lysis and PCR after laying $F_2$ progeny. From the first set of injections, a strain with the D1135L and S1136W substitutions was successfully isolated. Then, a second round of injections was made over this strain to introduce the remaining substitutions. However, the remaining two crRNAs were combined with tracrRNA and IDT Cas9 v3 nuclease at a final concentration of 2.1 μM to form RNPs in an attempt to increase the editing efficiency. Worms were injected with this injection mixture and genotyped as previously described[19]. The four remaining substitutions were successfully isolated and three independent lines were kept and frozen (CER658, CER659, and CER660).

**Indel mutation sequencing**. For indel examples in zebrafish (Supplementary Fig. 1), five embryos per injection were collected at 24 hpf and genomic DNA was extracted following a protocol adapted from Meeker et al., 2007[61]. Using this genomic DNA as template, a ~100 bp PCR product was obtained using the following parameters: 30 s at 98 °C, 35 cycles of [10 s at 98 °C, 30 s at 60 °C, and 30 s at 72 °C], and a final step at 72 °C for 2 min. In *C. elegans*, individual worms were collected ($F_2$ dumpies or $F_2$ wrmScarlet knockouts from $F_1$ heterozygotes) and genomic DNA was extracted via single worm lysis. Using this genomic DNA as template, a PCR product was amplified using a touchdown PCR program: 2 min at 98 °C, 11 cycles of [15 s at 98 °C, 15 s at 64 °C (decrease by 0.5 °C per cycle), and 30 s at 72 °C], 24 cycles of [15 s at 98 °C, 15 s at 59 °C, and 30 s at 72 °C], and a final step at 72 °C for 10 min. PCR products were visualized on agarose gel, purified (QIAquick PCR purification, Qiagen) and sequenced. After Sanger sequencing (Stab-vida), indel mutations were identified and analyzed using the ICE tool[62] (Synthego Co). Target sequences were aligned by mafft v7.271 using default options[63].

For mutagenesis quantification in the SpG and SpRY specific targets (Supplementary Fig. 7d), zebrafish embryos were injected in the optimized conditions (240 pg of gRNA and 300 pg of SpG/SpRY), collected 2 samples at 48 hpf and genomic DNA was extracted similarly as described above. Each sample contained 8 embryos as representation of the injection experiment. We amplified the region of interest from our mutants and WT controls (Supplementary Data 1) which were then Sanger sequenced. Then, we mainly used the ICE deconvolution tool[62] to calculate percentage of mutagenesis (for 3 gRNAs where ICE did not provide any output, we used the TIDE tool[64]). Quantified mutagenesis results are the average of the two collected samples per target (Supplementary Data 1 and Supplementary Fig. 7d).

Mutagenesis quantification in *C. elegans* was performed by injecting young adult hermaphrodites with the *dpy-10* gRNA and the co-injection markers *myo-2p::mCherry* and *myo-3p::mCherry*. mCherry-expressing $F_1$ animals were picked in pools of 10 animals before the appearance of Dpy or Rol phenotypes (earlier than L4 stage). Then, animals were phenotypically scored by the display of Dpy or Rol phenotype at a later time of development (later than L4). After phenotypic scoring, pooled animals were lysed and genotyped with primers that flank the region of interest (Supplementary Data 1), Sanger sequenced (Stab vida) and analyzed using the ICE tool[62]. Non-injected N2 animals were used as control.

**Statistics**. No statistical methods were used to predetermine sample size. The experiments were not randomized and investigators were not blinded to allocation during experiments and outcome assessment. Bar graphs show the average and are represented with S.E.M bars and violin plots are represented with individual data points and the median. Zebrafish phenotype or viability data in different injections conditions come from, at least, two independent injection experiments per figure panel. Zebrafish mutagenesis data per specific SpG and SpRY targets are coming from the average of 2 biological replicates. *C. elegans* data come from one or two independent experiments per figure panel with all data derived from parallel injections unless otherwise specified in the figure legend. Exact Fisher test, χ2-test (chi-squared), Student's *t* test, Mann–Whitney *U* test, Kruskal–Wallis, or one-way ANOVA test with Tukey's or Dunnett's tests for multiple comparisons were performed using SSPS 25.0 (IBM, Armonk, NY, USA), Prism (GraphPad Software v9, La Jolla, CA, USA), or R programming language. When applicable, all tests were two-sided. χ2-test of contingency was used to compare the results from different zebrafish injections and/or temperature incubation conditions (with Yates' correction for continuity when required).

**Reporting summary**. Further information on research design is available in the Nature Research Reporting Summary linked to this article.

## Data availability
Source data are provided with this paper. Key plasmids have been deposited in Addgene. Other relevant data are available within the article and its Supplementary Information files or from corresponding authors upon reasonable request.

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

## Acknowledgements

The authors thank Luis Hernandez-Huertas for his help with statistical analysis. We thank all members of the Moreno-Mateos and the Cerón laboratories for intellectual and technical support. Ceron laboratory thanks the CERCA Program (Generalitat de Catalunya) for their institutional support, and Nicholas Stroustrup and Alberto Villanueva for sharing reagents and equipment, and Marta Artal and Antonio Miranda for allowing to share the information on the efficiency of the *trx-1* and *usp-48* endogenous reporters generation. We thank Erik Jorgensen for sharing the EG9615 strain. We thank Jordan Ward for hosting J.C. at the UCSC where *cki-1* tagging was performed. This study at Cerón Lab has been funded by Ministerio de Ciencia, Innovación y Universidades, which is part of Agencia Estatal de Investigación (AEI), through the Retos grant, number PID2020-114986RB-100. This work was also supported by Ramon y Cajal (RyC-2017-23041) and MDM-2016-0687 programs (Spanish Ministerio de Ciencia, Innovación y Universidades), Universidad Pablo de Olavide (UPO) Research and the Springboard programs from UPO and CABD, respectively and has been co-funded by the Fondo Europeo de Desarrollo Regional (FEDER) and Consejería de Transformación Económica, Industria, Conocimiento y Universidades de la Junta de Andalucía, within the operative program FEDER Andalucía 2014-2020 (01 - Refuerzo de la investigación, el desarrollo tecnológico y la innovación, grant P20_00866) (M.A.M.-M.). The Moreno-Mateos lab is supported by PGC2018-097260-B-I00 grant from Spanish Ministerio de Ciencia, Innovación y Universidades and European Union. The CABD is an institution funded by Pablo de Olavide University, Consejo Superior de Investigaciones Científicas (CSIC), and Junta de Andalucía. C.S.-B. was a recipient of the Ayudas Puente Predoctorales (Universidad Pablo de Olavide, V Plan Propio de Investigación y transference). J.V. had an INPhINIT PhD fellowship from "la Caixa" Foundation (LCF/BQ/IN17/11620065) co-funded by Marie Skłodowska-Curie grant agreement no. 713673. D.B. is a predoctoral

fellow of the CONACYT "Becas al Extranjero" Program of Mexico. D.K. has an FI AGAUR fellowship (2018FI_B1_00511) from Generalitat de Catalunya. A.J.G. and B.P.K. acknowledge support from National Institutes of Health (NIH) R00 CA218870 and NIH P01 HL142494 (B.P.K.) and 5R35GM122580-05 (A.J.G.). Some figures in this manuscript were created with BioRender.com.

## Author contributions

M.A.M.-M. and J.C. conceived the project and designed the research. C.S.-B. generated SpG and SpRY plasmids for in vitro transcription and performed all zebrafish experiments with the contribution of I.M.-S., J.C.-C., and M.A.M.-M. Most *C. elegans* experiments were performed by J.V., with the contribution of J.C., D.B., D.K., M.R.-L., and M.C.-P. N.R.M. and C.C. purified SpG and SpRY recombinant proteins. C.E.V. and A.J.G. updated CRISPRscan and calculated number of genome targets with CRISPR scores. A.R. and A.J.P.-P. generated alignments of indel mutations and calculated the number of targets for different Cas enzymes in zebrafish and *C. elegans* genomes. M.A.M.-M., J.C., J.V., C.S.-B., and B.P.K. performed data analysis, and M.A.M.-M. and J.C. wrote the manuscript with input from J.V., C.S.-B., and B.P.K. All authors reviewed and approved the manuscript.

## Competing interests

The authors declare the following competing interests: B.P.K. is an inventor listed on U.S. Provisional Patent applications filed my Mass General Brigham that describe the development of the SpG and SpRY variants, including application 17/157,708 entitled "Unconstrained Genome Targeting With Near-PAMless Engineered CRISPR-Cas9 Variants" and application 17/157,805 entitled "CRISPR-Cas Enzymes With Enhanced On-Target Activity. B.P.K. is a consultant for Avectas Inc., EcoR1 capital, and ElevateBio, and is an advisor to Acrigen Biosciences, Life Edit Therapeutics and Prime Medicines. All the other authors declare no competing interests.
