## [Peer Review File · Nature Communications]

Reviewers' Comments:

Reviewer #2:

Remarks to the Author:

In this manuscript, Vicencio et al describe the optimization of SpCas9 variants, SpG and SpRY that recognizes NGH or NAN PAM sequences using zebrafish and *C. elegans* models. The authors performed multiple optimizations such as testing phenotype penetrance using Cas9 mRNA and protein and tested these variants on limited PAM sequences. These variants could expand the targeting sequences in zebrafish and *C. elegans* many fold, and will be useful to the community. However, data presented in this manuscript does not show how many PAMs can be targeted efficiently as only a limited number of loci are targeted and quantification of mutagenesis activities is not provided. More specific points are as follows:

Page 4, Figure 1a-d: Authors investigated the efficiencies of SpG and SpRY variants on NGG PAM sites. Do we expect these variants to retain WT activity? If no, then how did the authors conclude higher concentration of sgRNA/Cas9 is required to achieve higher mutagenesis rates? Would it be better to select the PAMs that work for these variants and then optimize?

A quantification of mutagenic activity is required, and perhaps testing more than one guide would provide more convincing data as to how effective these variants are in vivo.

Fig 1d. How did the authors decide the 300pg cas9 and 240 pg sgRNA concentrations are the highest doses to achieve maximum mutagenesis efficiency? Previously, Liu et al 2014 showed that 400 pg Cas9 provides ~80%, and 800 pg cas9 can produce 100% mutagenesis efficiency. Would you please provide the dosage curve and quantification data?

Also, in the manuscript, the authors state, "such increase did not affect embryonic viability, suggesting a lack of toxicity-associated effects", however, Supp Fig 2a shows increasing cas9 and sgRNA cause lower viability; please clarify.

How many guides and genes were tested with increased dose? Could it be gene/guide specific because only 1/2 guide has higher activities?

Please quantify the effect of temperature on mutagenesis activity? It is unclear whether any increase in phenotype penetrance in zebrafish at higher temperatures (28C vs 32C or 34C similar to Cas12a).

Figure 1f, please quantify the activity by sequencing to show the specificity of the variants.

Figure 1i What was the concentration of guide RNA? Any dosage curve? How did you arrive at 8uM of Cas9 protein being optimal?

Figure 2 15 targets with NGH PAM were tested; NGH PAM can generate 12 different combinations; how many combinations were tested? It looks like only 6/15 guides worked. It would be nice for the readers to know the most efficient PAMs targeted by SpG and/or SpRY? For clarity to readers, figure 2 can be summarized in a table. Quantification of mutagenesis activity will help understand the efficacy of these variants.

What do you mean by "12 out of 15 showed some activity", 2%, or 5% or 20%?

Page 6: " we identified DNA lesions induced by SpG activity that were similar to what is described for SpCas9 with short insertions or deletions", however, Supp Figure 5 only shows deletions; please label the figure to show the size of indels. Running these samples through next-generation sequencing can provide detailed mutagenesis signatures.

Page7: please elaborate what does some activity and high activity mean, any sequencing data?

Page 7: "Altogether, our results demonstrate that SpG and SpRY can efficiently generate mutants in zebrafish embryos at genomic sites where SpCas9 is poorly active or inactive, with variable

activity among different targets”, please elaborate what are these different targets, what are the most efficient PAM sequences for these variants in zebrafish?

Page 7: Please cite relevant reference(s) related to KCl concentration affect the editing efficiency.

Figure 3a Does the first or third base in NGH or NAN affect the editing efficiency?

Figure 3d: Please provide quantification of mutagenesis efficiency.

Page 8/9: Since CRISPRScan was developed based on a machine learning approach, how many targets were used to modify the algorithm for SpG and SpRY variants.

Data presented here do not clearly show a positive correlation. It is important given a recent preprint by Uribe-Salazar et al. showed poor co-relation between CRISPRScan score and mutagenesis score. A table summarizing the score and mutagenesis rate will be helpful to the readers.

Page 10: Did you sequence the inserts following HDR? What is the rate of correct insertion? How did you get 19.7% and 13.1% numbers? What percentage of progenies had GFP expression?

How does HDR rate by these variants compared with Cas9? What are the most efficient PAMs for HDR?

I think HDR experiments are done superficially and require more robust data to convince the readers.

Did authors try HDR in zebrafish?

Reviewer #3:

Remarks to the Author:

The manuscript by Vicencio et al describes the first use of Cas9 variants SpG and SpRY in vertebrate and invertebrate animal model systems. They show that these Cas9 variants are able edit DNA using noncanonical PAM sites. These findings are noteworthy and a significant advancement in the field of genome editing. The experiments performed are well controlled and the genome editing results albeit on previously characterized loci are clear. In this reviewers opinion this is a high quality manuscript that would suitable for publication provided the following minor changes are made.

In this reviewer’s opinion, further analysis of how SpG and SpRY enhance the genome editing toolbox for Zebra fish and *C. elegans* is warranted. For instance, supplemental Figure 1b provides the total number of targets per ORF, but it would be more helpful to know how many guides are available per open reading frame and how many of these have a CRISPRscan score of greater than 66. If all orfs are targetable with multiple guides with a score greater than 66 this would be important to note. In addition, the authors state that this will enable targeting of miRNAs and other short RNAs which is certainly a significant advancement over wtCas9. Analysis was not done to show how many of these miRNAs are not targetable with wtCas9 or what percentage of miRNAs would now be considered targetable with these new variants. For instance, what percentage of miRNA genes now have a target with a CRISPR scan score of greater than 66 and what percentage would have that score or greater with only wtCas9. This reviewer appreciates that predicting if something is targetable for editing is not accurate 100% of the time, and as such making broad statements such as a certain percentage of the genome is now accessible is often times not prudent. However, further analysis of how much better these Cas9 variants are to the wt will strengthen the argument that these results are a very significant advancement in the field.

In addition, other minor edits to clarify the text and results are listed below.

Page 4 Purified Cas9 proteins are His tagged for purification but His-tagged SpCas9 is referred to

as wild type in the manuscript.

Page 5 In the manuscript the authors state that proteins were tested in vitro at different temperatures including 15, 25, 37, and 50 C. This leads the reader to think other temperatures were tested. If they were please indicate the results. If they were not, please reword so that we know these are the only temperatures that were tested.

Page 6 Authors state that Cas9 can exhibit high editing activities in animals grown at different temperatures. While this is an accurate statement, the way it is worded leads the reader to think that animals of the same species were grown at different temperatures when in fact I think it was different species at different temperatures. If this is the case this should be rewritten to clearly reflect this.

Page 11 At the time of this review SpRY has also been successfully applied in fungi and this is not cited.

Page 12 Authors write "Finally, high-fidelity versions can decrease off-target activity as shown in mammalian cells." It is unclear what is meant by the addition of this statement. This reviewer assumes these experiments have not been done. Further explanation and discussion of how these high fidelity enzymes would be expected to change editing of off target activity is needed if this statement remains.

Dear Reviewers,

We would like to thank you for your insightful comments on the manuscript. We have addressed your suggestions that were mainly focused on providing mutagenesis data, additional HDR experiments in *C. elegans*, analysis of temperature control of SpG and SpRY and computational studies to determine the number of potentially efficient genomic targets for these two variants. All changes can be visualized through track changes. Please find below a point-by-point answer to your questions and suggestions:

REVIEWER COMMENTS

Reviewer #1 (Remarks to the Author):

In this manuscript, Vicencio et al describe the optimization of SpCas9 variants, SpG and SpRY that recognizes NGH or NAN PAM sequences using zebrafish and C. elegans models. The authors performed multiple optimizations such as testing phenotype penetrance using Cas9 mRNA and protein and tested these variants on limited PAM sequences. These variants could expand the targeting sequences in zebrafish and c. elegans many fold, and will be useful to the community. However, data presented in this manuscript does not show how many PAMs can be targeted efficiently as only a limited number of loci are targeted and quantification of mutagenesis activities is not provided.

We thank the reviewer's comments and suggestions and are glad to hear that she/he found the study useful to the community. Previous studies with SpG and SpRY in mammalian cell cultures and plants did not detect any strong bias in terms of efficiency among different possible PAMs (Wessels et al., 2020 Science, Zhang et al., 2021 NAR, Li et al., 2021 Mol. Plant, Ren et al., 2021 Nature Plants). Here, we have analyzed 25 and 18 targets for SpG and SpRY in zebrafish and *C. elegans*, respectively. Since SpG and SpRY shared NGN PAM targeting, we focused on NAN targets for SpRY. Between *C. elegans* and zebrafish, we have tested all NGN and NAN possible PAM combinations (regarding the 3rd nucleotide within the PAM) and detected mutants and activity in most of them. However, more targets would be needed to better predict the cut efficiency through particular PAMs with these new systems in animals.

In addition, the data mentioned above from mammalian cell culture and plants experiments strongly suggest that 3rd position in the PAM sequence is not a critical factor controlling SpG or SpRY activity for NGN or NAN targeting, respectively. Notably, target nucleotide composition bias and gRNA stability have been shown as important determinants that influence CRISPR-SpCas9 activity (Moreno-Mateos et al., 2015 Nature Methods, Doench et al., 2014 and 2016, Nature Biotech). Indeed, we found that an algorithm to predict CRISPR-SpCas9 activity *in vivo*, CRISPRscan (Moreno-Mateos et al., 2015), can significantly detect highly efficient targets in zebrafish for SpG and SpRY since they share similar gRNA scaffold sequence. Therefore, while more targets would be analyzed to determine whether the within and outside the PAM sequence may significantly influence SpG or SRY activity in animals, we and others have not observed a clear bias among different PAMs. We have clarified this point in the results section (page 9) as follows:

"Since SpG and SpRY are modified versions of SpCas9 and no bias have been observed in terms of PAM preferences for these Cas enzymes (Wessels et al., 2020, Zhang et al., 2021, Li et al., 2021, Ren et al., 2021), we tested whether the

CRISPRscan algorithm built for SpCas9 was also able to predict the activity of these minimal PAM nucleases in zebrafish.”

Regarding the quantification of mutagenesis, the phenotype penetrance correlates with CRISPR-Cas mutagenesis activity in our model systems. Such correlation has been observed in F0 zebrafish injected embryos (*Moreno-Mateos et al.*, 2015, *Moreno-Mateos et al.*, 2017, *Wu et al.*, 2018 *Kroll et al.*, 2021), and in the *C. elegans* germline (*Dokshin et al. 2018 Genetics*).

In any case, as the reviewer suggested, we have now included experiments quantifying mutagenesis activity in *C. elegans* and zebrafish that confirm the correlation between phenotype penetrance and CRISPR-Cas activity (Supplementary Fig. 3b-e and Supplementary Fig. 7d).

We have now modified the text as follow:

Page 5: “Then, since mutant phenotypes correlate well with CRISPR activity in *C. elegans* (*Dokshin et al.*, 2018, Supplementary Fig. 3b-e), we validated the tolerance to mismatches *in vivo* by scoring a dominant phenotype caused by *dpy-10* targeting, by using SpG with the matched guide and a guide with a mismatch at position +5”.

Page 7: Notably, as we observed in *C. elegans*, phenotype quantification using SpG and SpRY correlated with genome mutagenesis in zebrafish where highly efficient targets showed the highest indels levels (Supplementary Fig. 7d and Supplementary Table 1). Indeed, highly efficient targets showed an average of 68.5 % with a distribution from 40% to 91.5% (Supplementary Fig. 7d).

More specific points are as follows:

Page 4, Figure 1a-d: *Authors investigated the efficiencies of SpG and SpRY variants on NGG PAM sites. Do we expect these variants to retain WT activity? If no, then how did the authors conclude higher concentration of sgRNA/Cas9 is required to achieve higher mutagenesis rates? Would it be better to select the PAMs that work for these variants and then optimize?*

Yes, since they are variants derived from WT Cas9, it is expected that SpG and SpRY retain WT activity. In fact, these variants previously showed activity in NGG PAMs in mammalian cell culture (*Wilson et al. 2020*). Indeed, in our optimized conditions, we recapitulated similar results to those obtained in mammalian cell culture where Cas9 and SpG globally had quite comparable activity, and SpRY performed with slightly lower efficiency (*Wilson et al. 2020*). Moreover, we tested SpG and SpRY on NGG PAM targets because these are the only genomic sites shared with WT SpCas9. Using these NGG common targets, we realized the lower efficiency of SpG and SpRY at standard concentrations of gRNAs and mRNA normally used for Cas9 in zebrafish. Then, by increasing gRNA and mRNA concentration, we were able to detect high activity from SpG and SpRY.

A quantification of mutagenic activity is required, and perhaps testing more than one guide would provide more convincing data as to how effective these variants are in vivo.

As mentioned above, once established optimized injection conditions for SpG and SpRY systems, we tested 23 and 17 targets more in zebrafish and *C. elegans*, respectively. We have now quantified the efficiency of most of these gRNAs by

Sanger sequencing and ICE and TIDE deconvolution. This new data which is now in Supplementary Fig. 7d and Supplementary Fig. 3b-e showed that phenotype penetrance generally correlates with mutagenesis.

Fig 1d. *How did the authors decide the 300pg cas9 and 240 pg sgRNA concentrations are the highest doses to achieve maximum mutagenesis efficiency? Previously, Liu et al 2014 showed that 400 pg Cas9 provides ~80%, and 800 pg cas9 can produce 100% mutagenesis efficiency. Would you please provide the dosage curve and quantification data?*

We thank the reviewer for this suggestion. We did not try higher concentrations since at 300 pg of SpG and SpRY using 240 pg of gRNA we were already able to recapitulate the activity described for these new Cas9 versions in mammalian cell culture analyses (Wilson et al. 2020) where SpG showed comparable activity to Cas9 being slightly lower for SpRY. We have now used 600 pg of Cas mRNA and 480 pg of gRNA in our two target sites with NGG PAM sequences to be able to compare the activity of Cas9, SpG and SpRY side-by-side. While we did not observe a clear increased activity, we detected a decrease in the zebrafish embryo viability from 61-75% at 300 pg mRNA and 240 pg of gRNA to 51-56% when using SpG and SpRY at 600 pg and 480 pg of gRNA. Similar or even higher decrease in viability is also observed for Cas9. Moreover, increasing too much the amount of Cas mRNA and gRNA could imply a higher risk in terms of off-targets effects. Therefore, we believe that going beyond 300 pg of mRNA and 240 pg of gRNA does not provide any practical advantage. This new data is now part of Supplementary Figure 2, and it is mentioned in the text as follows (page 4):

“However, much higher concentrations of mRNA and gRNA decreased embryo viability while did not enhance SpG and SpRY activity, suggesting that an excess of these two components does not improve the system performance (Supplementary Fig. 2a,d,e).”

Also, in the manuscript, the authors state, “such increase did not affect embryonic viability, suggesting a lack of toxicity-associated effects”, however, Supp Fig 2a shows increasing cas9 and sgRNA cause lower viability; please clarify.

Thanks, the reviewer is correct, there is a slight decrease in the viability when using 300 pg of Cas9 and 240 of gRNA, but such effect is not statistically significant. However, and as stated above, increasing mRNA Cas and gRNA concentration produced too high toxicity.

We have modified the text as follows (page 4, 1st paragraph):

“Such increase did not significantly affect embryonic viability”

How many guides and genes were tested with increased dose? Could it be gene/guide specific because only 1/2 guide has higher activities?

In zebrafish, we tested 25 targets at 3 different loci, 2 in NGG, 15 in NGH and 8 in NAN sites using optimized conditions (300 pg mRNA and 240 of gRNA). We observed high activity and high phenotype penetrance in a fraction of targets (10 out 25) when using these optimized concentrations. This could be likely due to the intrinsic variable efficiency among gRNAs/targets sites observed in Cas9 for thousands of targets (Moreno-Mateos et al. 2015, Varshney et al. 2015 Genome Research). Although we observed that for some gRNAs an increase in their

concentration (240 pg per embryo) together with a higher Cas9 mRNA concentration (300 pg per embryo) can enhance their performance, it is expected to observe variable activity among gRNAs even on optimized conditions. We have shown that an algorithm (CRISPRscan) developed in similar conditions (mRNA and gRNA injection in zebrafish embryo at one-cell stage) can help to select highly efficient gRNAs

Please quantify the effect of temperature on mutagenesis activity? It is unclear whether any increase in phenotype penetrance in zebrafish at higher temperatures (28C vs 32C or 34C similar to Cas12a).

We thank the reviewer for this suggestion that has helped to show that temperature is not a variable controlling SpG and SpRY activity *in vivo* (Supplementary Fig. 7f). In the original manuscript, we did not observe any difference between for SpG or SpRY activity *in vitro* at 25 °C and 37 °C. We have now tested the activity of 11 gRNAs (6 and 5 gRNAs for SpG and SpRY respectively) in zebrafish embryos at 28 °C and 34 °C. We have not observed an increase in the activity for 10 out of 11 gRNAs tested, suggesting that SpG and SpRY have similar efficiencies at 28 °C and 34 °C. Only one gRNA targeting a NAN PAM sequence site showed clearly and significantly higher activity, but we think that this could be due to something specific related with the structure of this gRNA in particular. For example, a better gRNA conformation at higher temperature that could improve gRNA loading to SpRY and facilitate the targeting.

We mentioned this result on page 7, second paragraph:

“Finally, we tested whether temperature could affect SpG and SpRY activity *in vivo*. As shown in our *in vitro* experiments, we did not observe any substantial difference in the activity of most of the gRNAs used (10 out of 11, Supplementary Fig. 7f) with SpG and SpRY at 28°C and 34°C, which is the highest temperature that allows normal zebrafish development (*Moreno-Mateos et al., 2017*)”

Figure 1f, please quantify the activity by sequencing to show the specificity of the variants.

As mentioned above, the correlation between phenotype and CRISPR-Cas mutagenesis is well established in *C. elegans* (*Dokshin et al., 2018, Genetics*). Such correlation was further validated by our new experiment added as Supplementary Figures 3b-e.

In the previous version of the manuscript, we performed the experiment shown in panel 1g to validate *in vivo* the specificity observed for the variants *in vitro*. In that experiment, we used a gRNA with a mismatch at position +5 since it showed slight activity *in vitro* whereas a mismatch at position +1 did not show any activity.

Still, and related to figure 1f, we have now included an additional experiment. Using ICE (Interference of CRISPR Editing), we have quantified the mutagenic activity of SpG and SpRY at the highest concentration (8 µM) using gRNAs without mismatches and with a mismatch at the +5 position. Both phenotypic and mutagenic quantification confirm the specificity of the Cas9 variants. This experiment is now included as Supplementary Figure 4 a, b, and commented in the text, on page 5, as follows:

“In a separate experiment, we compared the activity of SpG and SpRY (Interference of CRISPR Editing, Synthego) with *dpy-10* guides, both matched and mismatched at the +5 position (from Fig. 1f) by scoring phenotypes (Supplementary Fig. 4a) and quantifying mutagenesis by ICE (Interference of CRISPR editing, Synthego, Supplementary Fig. 4b). We observed that a single mismatch almost completely abolished the *in vivo* activity of SpG and SpRY.”

Figure 1i What was the concentration of guide RNA? Any dosage curve? How did you arrive at 8 μ M of Cas9 protein being optimal?

We used equimolar concentrations of nuclease (Cas9, SpG or SpRY), tracrRNA and crRNA. We realized that this information was absent in the previous version of the manuscript, and it has been added in the legend and in Materials and methods as follows:

Legend:

Figure 1i. An anti-*wrmScarlet* guide with NGG PAM was complexed with SpCas9, SpG, or SpRY in equimolar concentrations to compare their *in vivo* efficiencies in *C. elegans*.

Methods (p.22):

Injection mixes were prepared by combining equimolar concentrations of Cas9 nuclease, tracrRNA, and crRNA; and incubated at 37 °C for 15 minutes.

The dosage curve for that experiment, 1.3 μ M, 3.7 μ M and 8 μ M is shown in Figure 3b. In a previous manuscript where we described the methodology Nested CRISPR (*Vicencio et al., 2019 Genetics*), we eventually used high concentrations of Cas9 (see Table 1 at *Vicencio et al., 2019*). Higher concentrations of RNPs were not tested since 8 μ M was the highest possible concentration that can be achieved in the injection mix given the stock concentration of the RNPs after protein purification. Since SpG was near the saturation point of efficacy at 8 μ M and Cas9 toxicity was suggested at high concentration (*Dokshin et al., 2018*), we considered 8 μ M as close to optimal.

This information was mentioned in the text, on page 6, as follows:

“In a previous study, we found that Cas9 concentrations can be raised six-fold without any apparent toxicity¹⁸. Thus, when we used SpG at a six-fold higher concentration (8 μ M in the injection mix) we observed an efficiency of *wrmScarlet* targeting similar to that of SpCas9 (Fig. 1i).”

Figure 2 15 targets with NGH PAM were tested; NGH PAM can generate 12 different combinations; how many combinations were tested? It looks like only 6/15 guides worked. It would be nice for the readers to know the most efficient PAMs targeted by SpG and/or SpRY? For clarity to readers, figure 2 can be summarized in a table. Quantification of mutagenesis activity will help understand the efficacy of these variants.

We have now quantified the activity of most of these targets (21 out of 23) using Sanger sequencing and mainly ICE software for mutagenesis deconvolution and added this information to the Supplementary Table 1 as reviewer requested. We included all targets, their PAM sequences, if they are highly efficient or not according

to phenotype quantification and finally the percentage of mutagenesis. We have tested all possible PAM combinations (counting zebrafish and *C. elegans* together) and almost all of them in zebrafish embryos. Unfortunately, we have not analyzed enough number of targets to detect a potential predisposition for the PAM sequence in any of the new Cas endonucleases, SpG or SpRY. For that, a much higher number of targets (thousands) would be needed as it was used to generate algorithms to predict nucleotide position bias in the targets (Moreno-Mateos *et al.*, 2015, Doench *et al.*, 2016 Nature Biotechnology). However, as mentioned above, the differences between target mutagenesis efficiencies are not likely due to a PAM bias as showed in mammalian cell culture. The variability between gRNA activity may be explained by the nucleotide positions in the target region as shown for SpCas9 in mammalian cell culture and *in vivo* systems (Moreno-Mateos *et al.*, 2015; Doench *et al.*, 2016). Notably, we found that the CRISPRscan algorithm to predict SpCas9 activity in zebrafish embryos can be also useful for predicting highly efficient gRNAs for SpG and SpRY.

What do you mean by “12 out of 15 showed some activity”, 2%, or 5% or 20%?

We meant targets where at least 10% of embryos are mosaic mutant. We have now quantified the mutagenesis percentage in these embryos, and they show a minimal activity of 2.5% of indel mutations and a maximal of 91.5% (Supplementary Table 1 and Supplementary Fig. 7d).

Page 6: “ we identified DNA lesions induced by SpG activity that were similar to what is described for SpCas9 with short insertions or deletions”, however, Supp Figure 5 only shows deletions; please label the figure to show the size of indels. Running these samples through next-generation sequencing can provide detailed mutagenesis signatures.

We appreciate the reviewer’s comment. We have properly indicated the kind of indel mutations now. We have sequenced most of the zebrafish target sites used in this study, and we have found that mutations (insertions and deletions) induced by SpG and SpRY nucleases are similar to those described for SpCas9 activity, as expected. We show examples for some targets of each animal model to illustrate this. We have used Sanger sequencing, and deconvoluted the information mainly using ICE software (see methods for details). This is now Supplementary Fig. 6.

Page7: please elaborate what does some activity and high activity mean, any sequencing data?

We apologize to the reviewer for the lack of clarity. Our criterion was initially based on zebrafish embryo phenotype.

High activity meant: at least 50% of the embryos presented severe (Class II or alb/gol severe mosaic) or extremely severe (Class III or alb/gol like) phenotypes.

Some activity meant: at least 10% of embryos are mosaic mutants.

To avoid any confusion, we have rephrased it as follows: “12 out of 15 targets showed at least 10% of embryos mosaic mutants”

We have now measured mutagenesis level in most (21 out of 23) of the specific SpG and SpRY targets and we have found a great correlation between high mutagenesis level and what we called high active targets (Supplementary Fig. 7d). As stated in methods, we have collected two samples per target site at 48 hpf. Each sample contained 8 embryos as a representation of the injection experiment. We amplified

the region of interest from our mutants and WT controls and sequenced them by Sanger. Then, we mainly used ICE deconvolution tool to calculate percentage of mutagenesis (for 3 gRNAs where ICE did not provide any output, we used TIDE deconvolution). Indeed, high active targets showed an average of 68.5% of mutagenesis with a range between 40% and 91.5%. Likewise, those targets that show some activity according to the phenotype classification but were not considered high active has an 8 % of mutagenesis as average with a range between 2.5% and 22.5%. Notably, those targets classified as non-active by phenotype quantification (WT like) did not show any substantial activity (0.12% average). These data consolidate that phenotype quantification for *golden*, *albino* and *no-tail* is a fast and reliable method to indirectly measure CRISPR-Cas activity as we and others have previously shown (*Moreno-Mateos et al.*, 2015, *Moreno-Mateos et al.*, 2017, *Wu et al.* 2018, *Kroll et al.*, 2021). All detailed mutagenesis information is now part of the Supplementary Table 1.

Page 7: "Altogether, our results demonstrate that SpG and SpRY can efficiently generate mutants in zebrafish embryos at genomic sites where SpCas9 is poorly active or inactive, with variable activity among different targets", please elaborate what are these different targets,

This is shown in Supplementary Fig 7e. By those different targets we mean all targets we have used that show variable activity.

what are the most efficient PAM sequences for these variants in zebrafish?

As mentioned above, there is not significant differences in mammalian cell culture data among different PAM sequences. Although we do not appreciate any bias either in animals, we do not have not enough data here to conclude anything significant. Still, according to mammalian cell culture and our data, we believe that differential activity between targets is mainly due to other factors beyond the variability of PAM sequences. We have now mentioned this in the manuscript (page 9): "Since SpG and SpRY are modified versions of SpCas9 and no bias have been observed in terms of PAM preferences for these Cas enzymes (*Wessels et al.*, 2020, *Zhang et al.*, 2021, *Li et al.*, 2021, *Ren et al.*, 2021)..."

Page 7: Please cite relevant reference(s) related to KCl concentration affect the editing efficiency.

The seminal CRISPR paper by Jinek et al in 2012 showed that *S. pyogenes* Cas9 requires magnesium for cleavage and its activity is sensitive to certain ions (*Jinek et al.*, Science 2012). Since the elution buffer for SpG and SpRY contains 200 mM KCl, we wanted to ensure that the high KCl concentration in the injection mixes associated with high volumes of SpG and SpRY was not a confounding factor for editing efficiency.

Low and high concentrations of KCl were compared since some RNP-based protocols for genome editing in *C. elegans* promote increasing the ionic strength of the injection mix through the outright addition of KCl to prevent aggregation of the RNP (*Paix et al.*, 2017, Methods; *Prior et al.*, 2017, G3) whereas other protocols do not require KCl supplementation (*Dokshin et al.*, 2018, Genetics; *Vicencio et al.*, 2019, Genetics; *Ghanta & Mello*, 2020, Genetics). Since similar editing efficiencies were obtained by varying the KCl concentration while keeping the RNP

concentration constant, there is no evidence to support that differences in KCl concentration are responsible for differences in editing efficiencies.

This information has been added to page 8 of the manuscript as follows:

Since increasing the volume of the nuclease in the injection mix is accompanied by a parallel increase in KCl concentration, we tested whether the latter has an effect on editing efficiency. In addition, some protocols deliberately increase the ionic strength of the injection mix to prevent RNP aggregation (*Paix et al.*, 2017, Methods; *Prior et al.*, 2017, G3) whereas others do not (*Dokshin et al.*, 2018, Genetics; *Vicencio et al.*, 2019; *Ghanta & Mello*, 2020, Genetics).

Figure 3a Does the first or third base in NGH or NAN affect the editing efficiency?

As mentioned above:

“...data from mammalian cell culture and plants experiments strongly suggest that 3rd position in the PAM sequence is not a critical factor controlling SpG or SpRY activity for NGH or NAN targeting, respectively.”

This is mentioned in the text on page 9:

“Since SpG and SpRY are modified versions of SpCas9 and no bias have been observed in terms of PAM preferences for these Cas enzymes (*Wessels et al.*, 2020, *Zhang et al.*, 2021, *Li et al.*, 2021, *Ren et al.*, 2021)...”

Figure 3d: Please provide quantification of mutagenesis efficiency.

As explained and demonstrated above, there is a correlation between phenotype and mutagenesis, and we have provided three experiments reinforcing this fact. In the case of Figure 3d, SpRY seems the only nuclease effective to cleavage the target NAN 2 *in vitro*, and we demonstrated in Figure 3e how SpRY is also efficient *in vivo*, and WT Cas9 remains inefficient *in vitro* as *in vivo*. This experiment places SpRY as alternative nuclease to edit NAN sites.

The target of this experiment is the fluorescent marker *wrmScarlet*. Dokshin et al. et al showed how TIDE (similar to ICE) correlates with the loss of fluorescence. Therefore, the percentage of *wrmScarlet* alleles edited can be calculated by scoring the lack of fluorescence in F2 animals since lack of fluorescence in F2 progeny means editing in the two alleles at F1.

Thus, the percentage of alleles edited has been added as an additional line at the bottom of the *wrmScarlet* editing graphs in Figure 1 and Figure 3. The figure legends have been updated to reflect this additional line of data (page 30):

“The percentage of edited alleles was calculated as $(2 \times \text{no. of homozygous } F_1 \text{ knockouts} + \text{no. of heterozygous } F_1 \text{ knockouts}) / 2 \times \text{total no. of } F_1 \text{ screened} \times 100$.”

Page 8/9: Since CRISPRScan was developed based on a machine learning approach, how many targets were used to modify the algorithm for SpG and SpRY variants.

CRISPRscan is a linear regression model to predict SpCas9 activity *in vivo* (*Moreno-Mateos et al.*, 2015) and we have not modified this algorithm now. We have adapted it to test whether it can still predict SpG and SpRY activity, nucleases

derived from Cas9. Here we show that, to some extent, CRISPRscan can help to select highly efficient targets.

Data presented here do not clearly show a positive correlation. It is important given a recent preprint by Uribe-Salazar et al. showed poor co-relation between CRISPRScan score and mutagenesis score. A table summarizing the score and mutagenesis rate will be helpful to the readers.

While the ability to predict SpG and SpRY efficiency *in vivo* will be more accurate when generating high-throughput data specifically for these systems, we have shown that CRISPRscan can differentiate between highly efficient gRNAs and medium or low efficiency in a significant manner (Fig. 4a). This can help to select high active gRNAs *in vivo*. On the other hand, *Moreno-Mateos et al., 2015* and other reports (*Hauessler et al., 2016* or data from *Varshney et al., 2015*) have independently shown that CRISPRscan can select highly efficient gRNAs for SpCas9 activity *in vivo*. While the accurate correlation between activity and predicted score is not always possible, the algorithm was able to select highly efficient gRNAs among different gRNAs, and such capacity was maintained for SpG and SpRY.

The Uribe-Salazar et al. preprint showed a significant (albeit lower than expected) correlation between predicted scores and quantified mutagenesis.

This lower correlation between CRISPRscan scores and data from the Uribe-Salazar et al. preprint may be due to the different approaches used. While mRNA SpCas9 and gRNAs were injected directly in the cell of one-cell stage zebrafish embryos in CRISPRscan, SpCas9 purified protein and crRNA-tracrRNA were injected in the yolk in the Uribe-Salazar et al. preprint. Indeed, differences in the experimental approaches when using CRISPR-SpCas9 can be crucial to predict the system activity (*Hauessler et al., 2016*). We have now added the quantified mutagenesis information in Supplementary Table 1.

Page 10: Did you sequence the inserts following HDR? What is the rate of correct insertion? How did you get 19.7% and 13.1% numbers? What percentage of progenies had GFP expression?

Yes, we sequenced several inserts of the expected size, and they all matched with the expected sequence. This is now indicated in the new Supplementary Table 2. When using ssDNA as donor template, the PCR screen was performed on F₁ progeny that were singled out based on the display of the Dpy or Rol phenotypes, indicative of successful *dpy-10(cn64)* co-editing. Similar to what occurred with WT SpCas9, we observed partial insertions for the screened loci (Supplementary Table 2).

The percentage of successful HDR editing was calculated by dividing the number of correct sequenced samples between the total number of PCR-screened *dpy-10* co-edited animals. When using dsDNA as donor template, no PCR screen was necessary since we scored correct insertion by observing fluorescence. In the new supplementary table, we detail the absolute numbers for each of the experiments and loci.

Regarding the GFP expression, we would like to clarify that the presented numbers were calculated for the insertion of the Step 1 fragment of fluorescent proteins by the Nested CRISPR methodology. The Step 2 insertions were performed with SpCas9 WT (with classical NGG PAM crRNAs), and thus, the percentages of efficiency and

scoring of GFP::H2B and wrmScarlet were not included in this study as they are irrelevant for the evaluation of the SpG and SpRY variants.

We have now clarified this in the manuscript as follows (page 11):

“Then, also using SpG, we inserted a partial wrmScarlet sequence at the C-terminal end of two genes (*usp-48* and *trx-1*) lacking an NGG PAM proximal to the stop codon (**Fig. 4c**). This corresponds to the Nested CRISPR strategy¹⁹, where the insertion of a short sequence (≤ 200 bp ssDNA as repair template; step 1) facilitates the insertion of a longer fragment, using dsDNA as a repair template, to complete the fluorescent protein sequence (step 2).”

How does HDR rate by these variants compared with Cas9? What are the most efficient PAMs for HDR?

I think HDR experiments are done superficially and require more robust data to convince the readers.

Publications on improving HDR rates are focused on modifying the cellular repair pathways (*Nambiar et al.*, Nat Comm 2019) or in optimizing the donor template (*Richardson et al.*, Nature Biotechnology 2016; *Ghanta et al.*, Elife 2021). To our understanding, despite the existence of many Cas9 variants, there are no variants that specifically affect HDR. Therefore, the main limitation seems to be the cutting efficiency. In results from independent projects, we found that when editing with Cas12a, a more distant nuclease, the HDR rate is comparable to Cas9 with the cutting efficiency being the main limiting factor (Unpublished *C. elegans* data, and *Moreno-Mateos et al.*, 2017).

Still, to provide more HDR data as requested by the reviewer, and knowing that the nature of the donor template can influence HDR, we have added HDR experiments using both ssDNA and dsDNA as donors since all HDR events showed in the previous version were performed using ssDNA as repair template.

First, we used a 200 bp ssDNA repair template to tag a gene with a degron sequence, but this time we used a plasmid encoding a fluorescent reporter (*Pmlc-1::GFP*) as a co-marker. We obtained 16.6% of HDR events among green animals, which is similar to the efficiency range (from 6.6% to 19.7%) obtained using SpG combined with *dpy-10* mutations as co-marker.

This is now mentioned in the text as follows (page 11):

“Also using ssDNA as donor template, we made a translational reporter for *cep-1*, using SpRY and the stop codon TAA as NAN PAM (efficiency of 7.1% among co-edited worms), and tagged *cki-1* with a degron sequence with a 200-bp ssDNA repair template (efficiency of 16.6% among co-edited worms) (Fig. 4c).”

Moreover, we used a 752 bp dsDNA fragment as a donor template to make a GFP reporter through the second step of Nested CRISPR (*Vicencio et al.*, 2019). An editing efficiency of 21.7% was achieved using a commercial Cas9 (IDT) at 1.58 μ M (Table 1, *Vicencio et al.*, 2019). In comparison, using SpG at 3.8 μ M with the same NGG PAM, we obtained an efficiency of 22.9%. This result, along with the lower efficiency (2%) obtained by SpRY with an NGG PAM, aligns with the rest of the study, consolidating the idea that increased concentrations of SpG and SpRY are necessary to reach the efficiency levels of Cas9 at an NGG PAM.

This data has now been included in Supplementary Table 2, and in the updated Figure 4c, and is also mentioned in the text (page 11) as follows:

“Finally, we used a 752 bp dsDNA fragment as a donor template to generate GFP reporters at the *gtbp-1* locus with SpG and SpRY through the second step of Nested CRISPR at an NGG PAM (Vicencio et al., 2019). An editing efficiency of 21.7% was achieved using a commercial Cas9 (IDT) at 1.58 μ M (Vicencio et al., 2019), similar to the 22.9% obtained by using SpG at 3.8 μ M (Fig. 4c).”

Thus, these experiments provide additional support for the use of these Cas9 variants for HDR experiments. In any case, even for SpCas9 WT users, further studies are needed to determine the conditions for increasing HDR events in a reliable manner.

Did authors try HDR in zebrafish?

The advantage of using SpG and SpRY is the increase in the number of targets in the genome due to their simpler PAM sequences compared to the wild type SpCas9 where these new variants are coming from (Supplementary Fig. 1b and 9). We focused on optimizing them in zebrafish embryos, and now other applications will be possible in much more targets, including HDR. Therefore, we did not try it but, as observed in *C. elegans*, would expect similar outcomes for SpCas9 and the new variants in their optimized conditions.

Reviewer #2 (Remarks to the Author):

*The manuscript by Vicencio et al describes the first use of Cas9 variants SpG and SpRY in vertebrate and invertebrate animal model systems. They show that these Cas9 variants are able edit DNA using noncanonical PAM sites. These findings are noteworthy and a significant advancement in the field of genome editing. The experiments performed are well controlled and the genome editing results albeit on previously characterized loci are clear. In this reviewer's opinion this is a high quality manuscript that **would suitable for publication provided the following minor changes are made.***

We thank the reviewer for his/her comments on our manuscript and recommendation for publication after minor changes.

In this reviewer's opinion, further analysis of how SpG and SpRY enhance the genome editing toolbox for Zebra fish and C. elegans is warranted. For instance, supplemental Figure 1b provides the total number of targets per ORF, but it would be more helpful to know how many guides are available per open reading frame and how many of these have a CRISPRscan score of greater than 66. If all orfs are targetable with multiple guides with a score greater than 66 this would be important to note. In addition, the authors state that this will enable targeting of miRNAs and other short RNAs which is certainly a significant advancement over wtCas9. Analysis was not done to show how many of these miRNAs are not targetable with wtCas9 or what percentage of miRNAs would now be considered targetable with these new variants. For instance, what percentage of miRNA genes now have a target with a CRISPR scan score of greater than 66 and what percentage would have that score or greater with only wtCas9.

We thank for this suggestion that is very useful to visualize predicted highly active targets for SpG and SpRY. We have now included a detailed analysis of gRNAs available per ORF with a CRISPRscan score greater than 66 side-by-side with total number of targets per ORF for easier comparison. It is now included in Supplementary Figure 9 where we have also added a similar analysis for miRNA genes (mature miRNA sequence). For all possible comparison SpRY and SpG significantly provide more total and highly efficient targets (CRISPRscan score >66) in both zebrafish and C. elegans than SpCas9 with the only exception of SpG sites in mature miRNA loci where the differences were not significant (Mann Whitney U tests).

This reviewer appreciates that predicting if something is targetable for editing is not accurate 100% of the time, and as such making broad statements such as a certain percentage of the genome is now accessible is often times not prudent. However, further analysis of how much better these Cas9 variants are to the wt will strengthen the argument that these results are a very significant advancement in the field.

We agree with the reviewer that prediction are not 100% accuracy and we have changed the title of the results section about CRISPRscan as "CRISPRscan can help to predict SpG and SpRY highly efficient targets" from "CRISPRscan can predicts SpG and SpRY efficiency". However, our data from Supplementary Figure 9 showing how we have now much more available targets (including those

predicted to be highly efficient) per ORF and small elements such as mature miRNA loci indicate that with SpG and SpRY we have more targetable genomes. Our analysis here indicates that in the optimized conditions SpG and SpRY perform in a comparable manner than SpCas9 in animals (as it has been shown in mammalian cell culture and plants) and using these conditions we can have much more possibilities to edit the genome. In any case, we do agree with the reviewer that this is the first manuscript showing near PAM-less Cas functioning in animals and further studies with much more targets will clarify until what extent the activity of these variants is 100% comparable to SpCas9.

In addition, other minor edits to clarify the text and results are listed below.

Page 4 Purified Cas9 proteins are His tagged for purification but His-tagged SpCas9 is referred to as wild type in the manuscript.

Thanks for the correction. We have now mentioned in the main text (page 4) that all purified Cas (WT, SpG and SpRY) are His tagged.

Page 5 In the manuscript the authors state that proteins were tested in vitro at different temperatures including 15, 25, 37, and 50 C. This leads the reader to think other temperatures were tested. If they were please indicate the results. If they were not, please reword so that we know these are the only temperatures that were tested.

The reviewer is correct, it was confusing as it was written. We have now stated: "We tested these proteins in vitro at different temperatures: 15, 25, 37, and 50 °C" removing "including" that was the misleading term.

Page 6 Authors state that Cas9 can exhibit high editing activities in animals grown at different temperatures. While this is an accurate statement, the way it is worded leads the reader to think that animals of the same species were grown at different temperatures when in fact I think it was different species at different temperatures. If this is the case this should be rewritten to clearly reflect this.

We thank the reviewer for this. We have rewritten it as it follows: "...high editing activities in distinct animal species grown at different temperatures"

Page 11 At the time of this review SpRY has also been successfully applied in fungi and this is not cited.

We assumed that reviewer means this paper using SpRY in Dictyostelium that is an amoeba (PMCID: PMC8159936). We have added this reference in the discussion, page 12.

Page 12 Authors write "Finally, high-fidelity versions can decrease off-target activity as shown in mammalian cells." It is unclear what is meant by the addition of this statement. This reviewer assumes these experiments have not been done. Further explanation and discussion of how these high fidelity enzymes would be expected to change editing of off target activity is needed if this statement remains.

We have detailed what we meant here as follows: "Finally, high-fidelity versions can help to decrease off-target activity *in vivo* as shown in mammalian cells where

an increased in the specificity with only a slight reduction in the on-target activity was observed”.

Reviewers' Comments:

Reviewer #2:

Remarks to the Author:

The authors have addressed many of the minor points satisfactorily. However, the major point remains unanswered, and the authors did not perform extensive PAM analysis to make a conclusion presented in the paper. The premise of this paper is to broaden the targeting scope by using spG and spRY variants, and the authors claim to show editing in animals with minimal PAMs. They cited cell culture and plant data, but many PAMs which are successful in plants and cell culture did not show any activities in zebrafish. Therefore, cell culture and plant data cannot be extrapolated to the zebrafish and further characterization is warranted. The authors used only three loci and 23 targets to conclude that minimal PAM can be used, however, the data presented do not support their claims.

I analyzed the data presented in the supplementary table 1 based on their mutagenesis activities:

5/23 (GGT, CAA, CAG, AGT, AGC) targets showed no activities.

7/23 (TGT, CGT, TGC, GAG, CGT, TAT) targets showed activities less than 10%.

7/23 (AGG, CAG, ACG, CGC, GGC, GAA) targets showed activities more than 50%

4/23 (AAG, TGA, CAT, CGA) targets showed activities ranging from 10-50%

In conclusion, 50% of targets showed minimal activities and cannot be used to conclude that these variants can broaden the targeting coverage in zebrafish. Most PAMs were tested only once. Authors claim to have activities on NAA PAM, but only one out of two targets worked. In conclusion, the data presented here is very limited and requires further testing. The authors agreed in their rebuttal that more PAM testing is required.

Authors claim that spG and spRY shared NGN PAMs therefore they focused on NAN PAM, but they only tested NGN PAMs with spG enzyme.

There isn't any correlation between the mutagenesis activities predicted by CRISPRscan and the observed activity, and the data is pretty random to claim that CRISPRScan accurately predicts the spG and spRY activities.

Reviewer #3:

Remarks to the Author:

A fungal reference is PMC8265644. Otherwise all of my concerns were addressed.

Reviewer #2 (Remarks to the Author):

The authors have addressed many of the minor points satisfactorily. However, the major point remains unanswered, and the authors did not perform extensive PAM analysis to make a conclusion presented in the paper. The premise of this paper is to broaden the targeting scope by using spG and spRY variants, and the authors claim to show editing in animals with minimal PAMs. They cited cell culture and plant data, but many PAMs which are successful in plants and cell culture did not show any activities in zebrafish. Therefore, cell culture and plant data cannot be extrapolated to the zebrafish and further characterization is warranted. The authors used only three loci and 23 targets to conclude that minimal PAM can be used, however, the data presented do not support their claims.

First, we want to highlight that your suggestions in the first revision have been beneficial to improve the manuscript, and we appreciate your help. However, we courteously disagree that those points (about 20) addressed satisfactorily were minor. They entailed experimental effort in the last few months, and we believe that the new results and clarifications made a better manuscript.

We are convinced that an “extensive PAM analysis” is not necessary to support the conclusions of this paper. Our main conclusions are:

- 1) SpG and SpRY can be used to edit previously inaccessible sites in animal genomes.
- 2) A nuclease concentration higher than that used with Cas9 would be required for SpG/SpRY to obtain similar efficiency *in vivo* (probably due to a more thorough target scanning)
- 3) SpG and SpRY are sensitive to mismatches in the protospacer (which can have substantial implications on off-target effects).
- 4) SpG and SpRY can generate knock-ins by homology-directed repair.
- 5) RNP and mRNA-gRNA can efficiently result in SpG and SpRY activity.
- 6) *C. elegans* transgenics endogenously expressing SpG are viable and functional
- 7) CRISPRscan would provide some insights into the efficiency of editing with SpG and SpRY but more events need to be considered to make more precise predictions.

Thus, our point 7 agrees with the reviewer, but this does not affect the previous six conclusions. In fact, on page 13, we mentioned:

“Although more studies will be needed to further consolidate our data, we have updated our web tool (www.crisprscan.org) with the targets and predicted on- and off-targets scores for SpG (NGN) and SpRY (NRN)...”

Again, we agree with the reviewer that a much higher number of targets and loci would be necessary to better predict editing efficiency with SpG and SpRY. However, this is not within the scope of this study, and a bunch of additional targeted sites would still not facilitate a precise prediction of the activity (see below for more details on this). We suggested that CRISPRscan could help select highly efficient targets for SpG and SpRY, but we agree that we may soften the tone of our statements about the correlation between predictions and activities. Accordingly, we have modified the following sentences in the manuscript (changes highlighted in yellow):

Abstract:

Previous:

We have characterized and optimized the activity of SpG and SpRY in zebrafish and *C. elegans*.

Updated:

We have studied the nuclease activity of SpG and SpRY by targeting 40 sites in zebrafish and *C. elegans*.

Previous:

“We also found that the CRISPRscan algorithm can predict SpG and SpRY activity *in vivo*.”

Updated:

“We also found that the CRISPRscan algorithm could help to predict SpG and SpRY targets with high activity *in vivo*”

Page 9:

Previous:

“... we tested whether the CRISPRscan algorithm built for SpCas9 was also able to predict the activity of these minimal PAM nucleases in zebrafish.”

Updated:

“...we tested whether the CRISPRscan algorithm built for SpCas9 would help to predict the activity of these minimal PAM nucleases in zebrafish.

Page 10

Previous:

“Altogether, these results show that CRISPRscan predictions for SpG and SpRY efficiency can facilitate the use of these nucleases in zebrafish and demonstrate that RNP formulations can enhance editing activity particularly when poorly stable gRNAs are used.”

Updated:

“Altogether, these results show that CRISPRscan predictions for SpG and SpRY highly efficient targets could facilitate the use of these nucleases in zebrafish and demonstrate that RNP formulations can enhance editing activity particularly when poorly stable gRNAs are used.”

Page 13

Previous:

“Indeed, we have shown that CRISPRscan, an algorithm for predicting CRISPR-SpCas9 activity *in vivo*, can significantly facilitate the selection of highly efficient targets for SpG and SpRY.”

Updated:

“Indeed, we have shown that CRISPRscan, an algorithm for predicting CRISPR-SpCas9 activity *in vivo*, would facilitate the selection of highly efficient targets for SpG and SpRY.”

Previous:

“Although, more studies will be needed to further consolidate our data, we have updated our web tool (www.crisprscan.org) with the targets and predicted on- and off-targets scores for SpG (NGN) and SpRY (NRN) that will help in selecting the most active gRNAs.”

Updated:

“Since our study is limited to 40 targeted sites (25 in zebrafish and 15 in *C. elegans*), more studies will be needed to further evaluate specific predictive tools for SpG and SpRY activities. Meanwhile, we have updated our web tool (www.crisprscan.org) with the targets and predicted on- and off-targets scores for SpG (NGN) and SpRY (NRN).”

In any case, we would like to comment on the number of edited sites in our study.

It came to our attention that targeted sites in *C. elegans* were not considered. In fact, we have tested 40 sites in our study using SpG and/or SpRY (25 in zebrafish and 15 in worms), **covering and showing DNA editing activity *in vivo* in all PAM defined as optimal for SpG and SpRY editing**, NGA/T/C/G and NAA/T/C/G, respectively (Walton *et al.*, 2020) and using at least three sites per PAM except for NAT where we used two. This is in the line with recent publications showing SpG or SpRY editing in rice (Xu *et al.*, 2021 *Genome Biol*; Li *et al.*, 2021 *Mol. Plant*, Ren *et al.*; *Nature Plants* 2021) where a similar number of genomic targets with NGN or NAN PAM were tested (between 32 and 48). In addition, to further optimize these systems in animals, we edited most of our 40 sites with different concentrations of nucleases, temperatures or salt conditions, using RNP and mRNA-gRNA delivery approaches, and with and without mismatches on the protospacer.

Regarding the influence of the 1st or 3rd position in the PAM, the first study on SpG and SpRY led by our collaborator Dr. Kleinstiver clearly showed that there is no nucleotide composition bias in the 1st or 3rd position in the PAM that significantly influences SpG/SpRY activity, neither *in vitro* nor in human cells (Walton *et al.*, 2020.). However, they observed variable activity between target sites, with an average of 50% of mutagenesis for SpG in NGN sites and 30% for SpRY in NAN sites. After that, several papers in rice came out using a comparable number of targets as to what we have used in animals, with similar results: no clear bias in the 3rd position of the PAM and a variable activity among targets. In our study, we have consolidated the results from *in vitro*, human cells, and rice studies by editing 40 targets in two different animals. In any case, an important misconception here would be to think that editing efficacy relies only on the central three nucleotides of the PAM sequence. It has been demonstrated that nucleotides downstream, and especially upstream (protospacer region) are crucial in modulating the activity of every single CRISPR-Cas system. Determining the influence of the PAM surrounding nucleotides for a Cas system requires scoring hundreds or thousands of events, as, for example, was performed by Gasiunas *et al.* (Nat Comm. 2020) with novel Cas9 orthologs using randomized PAM libraries and sequencing. In addition, in eukaryotic genomes, other aspects such as the genomic context may also influence the efficiency. Indeed, initial studies on different CRISPR-Cas systems have been unable to demonstrate a bias within the N nucleotide of defined PAMs such as in SpCas9 (NGG, 2013) or As/LbCas12a (TTTN, 2015). Later studies using **thousands** of gRNAs were able to show some bias in the N of the SpCas9 PAM that could affect the final activity, together with other nucleotide biases on the protospacer and adjacent regions (Doench JG *et al.*, 2014 *Nature Biotech.*, Moreno-Mateos MA *et al.*, 2015 *Nature Methods*, Doench JG *et al.*, 2016 *Nature Biotech.*), or even redefine As/LbCas12a PAM to TTTV (Kim HK *et al.*, 2017 *Nature Methods*, Kim HK *et al.*, 2018 *Nature Biotech.*). Therefore, to test all these PAM combinations in a significant manner for ultimately improving activity prediction is **out**

of the scope of this manuscript but we believe that the field would cover that information in the future.

I analyzed the data presented in the supplementary table 1 based on their mutagenic activities:

5/23 (GGT, CAA, CAG, AGT, AGC) targets showed no activities.

7/23 (TGT, CGT, TGC, GAG, CGT, TAT) targets showed activities less than 10%.

7/23 (AGG, CAG, ACG, CGC, GGC, GAA) targets showed activities more than 50%

4/23 (AAG, TGA, CAT, CGA) targets showed activities ranging from 10-50%

In conclusion, 50% of targets showed minimal activities and cannot be used to conclude that these variants can broaden the targeting coverage in zebrafish.

As mentioned above, mutagenic activity does not depend only on PAM composition, and the 15 edited sites in *C. elegans* were not considered by the reviewer in this analysis. In any case, this is not a precise analysis of the global results in zebrafish:

First, among the 5 targets that the reviewer claimed no activity, there are 2 that were not quantified due to technical issues. To demonstrate the correlation between phenotype penetrance and mutagenesis level, we did quantify 21 out of 23 targets with NHG or NAN PAM and show that phenotype penetrance is a correct approach for evaluating target activity (Supp. Figure 7d). Notably, one (NGT PAM) out of these two targets without mutagenesis data showed that 50% of embryos had a mosaic phenotype, which strongly suggests that it is certainly an active target.

Second, among the targets with less than 10-15% of mutagenesis, 3 targets showed an increased activity (medium-high according to phenotype penetrance) when using RNPs: golden 3, golden 6, and no-tail 8. In addition, although temperature generally does not seem to influence SpG and SpRY efficiency, the target activity of no-tail 6 was rescued by increasing temperature, likely due to better gRNA stability or secondary structure conformation. Therefore, using different optimizations, we increased the activity of 4 additional targets.

Then, only 7 and not 11, out of the 23 NGH or NAN targets showed low activity when analyzed by mutagenesis levels and/or phenotype penetrance. Therefore, 6 out of 8 in SpRY, and 10 out of 15 in SpG showed medium-high activity under optimized conditions defined in our manuscript. Finally, the quantification of mutagenesis has showed 36% and 26.5% of mutagenesis average for SpG and SpRY, respectively. These results are comparable to human cells (Walton et al. 2020) and rice data (Li et al., 2021, Ren et al., 2021 Nature Plants) with 50-40% and 30-20% mutagenesis for SpG and SpRY, respectively and where targets showing low activity were also detected as observed for other CRISPR-Cas systems.

Most PAMs were tested only once. Authors claim to have activities on NAA PAM, but only one out of two targets worked. In conclusion, the data presented here is very limited and requires further testing.

We respectfully think that this statement is not completely accurate. Using *C. elegans* and zebrafish, we have tested all PAM defined as optimal for SpG and SpRY (NGA/T/C/G and

NAA/T/C/G, respectively) in at least three different sites except for NAT where we did it in two targets and, indeed, we have observed activity in each one of these PAMs. In particular, we also tested and show activity in all PAMs in zebrafish except for NAC which was tested in 3 targets in *C. elegans* (2 of them showing clear activity, Figure 3e). Therefore, with our 40 analyzed targets, we have provided enough data to state that efficient genome editing in animals with minimal PAM CRISPR-Cas9 enzymes is possible.

The authors agreed in their rebuttal that more PAM testing is required.

Yes, as mentioned above, we agree that more PAM testing is required for superior activity prediction, but again, this is not the main aim of this study. We referred to the fact that more targets would be needed to better predict the cutting efficiency in particular PAMs using these new systems in animals. To detect potential predisposition (unlikely with current data in different systems) for the PAM sequence in any of the new Cas endonucleases, SpG or SpRY, a much higher number of targets (hundreds or even thousands) would be needed, similar to what has been done to generate computational models for predicting nucleotide position bias in other CRISPR-Cas systems (Doench JG et al., 2014, Doench JG et al., 2016, Moreno-Mateos MA et al., 2015 and Kim HK et al., 2017 and 2018). To make this statement clearer, we have modified a sentence in the discussion as follows:

Page 13

Since our study is limited to 40 sites (25 in zebrafish and 15 in *C. elegans*), more studies will be needed to further evaluate predictive tools for SpG and SpRY activity. Meanwhile, we have updated our web tool (www.crisprscan.org) with the targets and predicted on- and off-targets scores for SpG (NGN) and SpRY (NRN).

However, our data together with *in vitro*, human cell, and rice data do not suggest a clear PAM bias and points to the fact that differential activity among different targets is likely due to the protospacer sequence and the genomic context. This is certainly something to be expected with SpG and SpRY, since other Cas nucleases do not also show activities of 100% in every single target site. In any case, we compared SpG and SpRY with Cas9 and show that by increasing the concentration of the variants, we can reach efficiency levels similar to that of Cas9.

In summary, beyond zebrafish data, half of the manuscript is also dedicated to *C. elegans*, and the complete analysis in both organisms using SpG and SpRY would motivate people to use these nucleases that we are aware is currently happening after our preprint was released in June.

Authors claim that spG and spRY shared NGN PAMs therefore they focused on NAN PAM, but they only tested NGN PAMs with spG enzyme.

In Figure 3C, we edit an NGC site with SpG and SpRY in *C. elegans* and SpG showed more activity than SpRY in that target. Similarly, SpG is noticeably more active than SpRY in NGH according to human cell data (Walton et al., 2020) and we initially observed similar results in NGG targets where, under our optimized conditions, SpG performed always equally or better than SpRY (Figure 1 and Supp Figures 2-4). Therefore, we

focused on the analysis and optimization of SpRY in NAN sites where SpG and SpCas9 have low efficiency or are inactive. Besides that, SpG will have less potential off-targets effects (by chance) which is another reason to use it in NGH sites instead of SpRY.

There isn't any correlation between the mutagenesis activities predicted by CRISPRscan and the observed activity, and the data is pretty random to claim that CRISPRScan accurately predicts the spG and spRY activities.

We respectfully disagree with the reviewer on this comment since the data is not random. We have used mutagenesis quantification to demonstrate a correlation between high level of activity and high penetrance phenotypes (Supp. Figure 7d). Using all zebrafish data based on phenotype penetrance, we observed a significant trend (Figure 4a, Fisher test) that suggests this computational model could help select highly efficient targets. We think that this can be useful for the community. For that reason, we have also generated a new figure, suggested by the other reviewer, showing the number of total targets of SpG and SpRY and those that are predicted to be highly efficient in both animal systems (Supp. Figure 9). In addition, we have updated the CRISPRscan website to provide this information to the users.

We agree with the reviewer that there is no 100% accurate correlation between activities and predicted scores. However, after our data analysis based on 25 targets, we believe that the algorithm can help differentiate between highly active targets and others with lower efficiency. Again, we agree that the model does not accurately predict SpG and SpRY activity *in vivo*, and to clarify this, we have softened the tone of a few sentences (see above changes in abstract, page 9,10, and 13).

Reviewer #3 (Remarks to the Author):

A fungal reference is PMC8265644. Otherwise all of my concerns were addressed.

Thanks for your helpful review. The reference has been corrected now.

Note from authors:

We have detected two small mistakes in Figures 2 and 4. They were errors in the figure composition, but the related concepts were right and clear in the main text.

Thus, in panel 2b, we set the SpG instead of SpRY.

In panel 4c, graphical legends for editing in *cki-1* and *cep-1* were swapped but correctly explained in the main text and supplementary tables.

Reviewers' Comments:

Reviewer #2:

Remarks to the Author:

I did not overlook c.elegans data. I could only find mutagenesis quantification for zebra fish in the attached excel spreadsheet. I tried to look for c.elegans target mutagenesis efficiency elsewhere in the manuscript but could not find it.

I looked into C.elegans crna tab, where target information is given. I could only find one target with TAA PAM and nine other targets covering NAN, NGN PAMs.

Based on the limited data presented in this manuscript (both c.elegans and zebra fish). I am not convinced there is enough data to claim flexible PAMs able to edit the genome efficiently, at best selected PAMs.

If authors claim otherwise, more robust PAM data with mutagenesis efficiency is required.

RESPONSE TO REVIEWERS, third revision, NCOMMS-21-21038A

Reviewer #2 (Remarks to the Author):

I did not overlook c.elegans data. I could only find mutagenesis quantification for zebra fish in the attached excel spreadsheet. I tried to look for c.elegans target mutagenesis efficiency elsewhere in the manuscript but could not find it.

Author Response: In our first revision of our manuscript submitted in December, we addressed the quantification of the mutagenesis efficiency in *C. elegans* in different ways, including ICE (Interference of CRISPR Editing) analyses. The data can be found:

- at the bottom of the *wrmScarlet* editing graphs in Figure 1 and Figure 3
- in Supplementary Figures 3B-E, using ICE analyses to demonstrate the correlation between phenotype and mutagenesis in our system
- in Supplementary Figure 4B

These are fragments of our response to our first revision related to this point:

Previous Reviewer #2 Comment: Figure 3d: Please provide quantification of mutagenesis efficiency.

Previous Author Response: As explained and demonstrated above, there is a correlation between phenotype and mutagenesis, and we have provided three experiments reinforcing this fact. In the case of Figure 3d, SpRY seems the only nuclease effective to cleavage the target NAN 2 in vitro, and we demonstrated in Figure 3e how SpRY is also efficient in vivo, and WT Cas9 remains inefficient in vitro as in vivo. This experiment places SpRY as alternative nuclease to edit NAN sites. The target of this experiment is the fluorescent marker *wrmScarlet*. Dokshin et al. et al showed how TIDE (similar to ICE) correlates with the loss of fluorescence. Therefore, the percentage of *wrmScarlet* alleles edited can be calculated by scoring the lack of fluorescence in F2 animals since lack of fluorescence in F2 progeny means editing in the two alleles at F1. Thus, the percentage of alleles edited has been added as an additional line at the bottom of the *wrmScarlet* editing graphs in Figure 1 and Figure 3. The figure legends have been updated to reflect this additional line of data (page 30):

The percentage of edited alleles was calculated as $(2 \times \text{no. of homozygous F1 knockouts} + \text{no. of heterozygous F1 knockouts}) / 2 \times \text{total no. of F1 screened} \times 100$.

Previous Reviewer #2 Comment: Figure 1f, please quantify the activity by sequencing to show the specificity of the variants.

Previous Author Response: "As mentioned above, the correlation between phenotype and CRISPR-Cas mutagenesis is well established in *C. elegans* (Dokshin et al., 2018, Genetics). Such correlation was further validated by our new experiment added as Supplementary Figures 3b-e.

In the previous version of the manuscript, we performed the experiment shown in panel 1g to validate in vivo the specificity observed for the variants in vitro. In that experiment, we used a gRNA with a mismatch at position +5 since it showed slight activity in vitro whereas a mismatch at position +1 did not show any activity.

Still, and related to figure 1f, we have now included an additional experiment. Using ICE (Interference of CRISPR Editing), we have quantified the mutagenic activity of SpG and SpRY at the highest concentration (8 μM) using gRNAs without mismatches and with a mismatch at the +5 position. Both phenotypic and mutagenic quantification

confirm the specificity of the Cas9 variants. This experiment is now included as Supplementary Figure 4 a, b, and commented in the text, on page 5, as follows:

"In a separate experiment, we compared the activity of SpG and SpRY (Interference of CRISPR Editing, Synthego) with *dpy-10* guides, both matched and mismatched at the +5 position (from Fig. 1f) by scoring phenotypes (Supplementary Fig. 4a) and quantifying mutagenesis by ICE (Interference of CRISPR editing, Synthego, Supplementary Fig. 4b). We observed that a single mismatch almost completely abolished the in vivo activity of SpG and SpRY."

Previous Reviewer #2 Comment: A quantification of mutagenic activity is required, and perhaps testing more than one guide would provide more convincing data as to how effective these variants are in vivo.

Previous Author Response: As mentioned above, once established optimized injection conditions for SpG and SpRY systems, we tested 23 and 17 targets more in zebrafish and *C. elegans*, respectively. We have now quantified the efficiency of most of these gRNAs by Sanger sequencing and ICE and TIDE deconvolution. This new data which is now in Supplementary Fig. 7d and Supplementary Fig. 3b-e showed that phenotype penetrance generally correlates with mutagenesis.

Reviewer #2 Comment: I looked into *C. elegans* crna tab, where target information is given. I could only find one target with TAA PAM and nine other targets covering NAN, NGN PAMs. Based on the limited data presented in this manuscript (both *C. elegans* and zebra fish). I am not convinced there is enough data to claim flexible PAMs able to edit the genome efficiently, at best selected PAMs. If authors claim otherwise, more robust PAM data with mutagenesis efficiency is required.

Author Response: As mentioned in your previous response, we have addressed all minor points satisfactorily. However, you insisted that we did not perform an extensive PAM analysis. The PAM requirements of SpG and SpRY have been extensively characterized in vitro and in human cells, results that have been supported now by many other studies and extended to other organisms as well (i.e. plants, bacteria, etc.). Although we agree that a model to predict editing efficiency with the SpG and SpRY variants would be interesting, we previously argued that hundreds or thousands of targets would be needed to produce a reliable predictive tool. In fact, no such tool exists even for human cells. In our animal models, particularly in *C. elegans*, such numbers would be reachable only via a concerted community effort but not in a single study. In yeast, in a recent article published in Nature Communications (<https://www.nature.com/articles/s41467-022-28540-0>), authors used more than 7000 targets to develop a predictive tool for SpCas9 and LbCas12a. Still, they mention that the predictive tool "had limited ability to accurately predict low-activity guides for Cas9". Thus, we maintain our view that although this is an interesting and useful suggestion, it falls far beyond the scope of our present study.

The main point of our current manuscript is to demonstrate, for the first time in animals, that SpG and spRY can edit NGH and NAN sites that cannot be edited with SpCas9. We showed this in vivo in seven targets in figures 3E, 4D, S3B, and S7. Of course, as it occurs with other nucleases, some sites will be more challenging to edit than others, but our study would encourage other scientists to use SpG or SpRY on sites that cannot be reachable by WT Cas9.

We edited about 40 sites in the *C. elegans* and zebrafish genomes. Instead of increasing the number of edited sites, our approach to supporting the use of SpG and SpRY in animals was to test different conditions: ex. Different concentrations of nucleases, different temperatures, different concentrations of salts, or RNPs vs. mRNA. All these different conditions are shown in Figures 1D, 1I, 3B, 3C, 3E, 4B, S2A, S2B, S2C, S2D, S2E, S3B, S3C, S5A, S5B, S5C, S7A, S7B, S7C, S7F, S8A, S8B, S10A, S10B, and S10C. Thus, although we targeted "only" forty sites, we have counted about 200 editing experiments. Such a strategy was essential to find clues about the editing of NGH and NAN PAM sites, such as the higher concentration required for SpG and SpRY compared to WT Cas9, or the sensitivity of these variants to mismatches.

As stated in our previous response, our study led to several conclusions that are supported by the data. However, again, making a predictive tool was not one of the aims of this study. We understand that the first version of the manuscript could have been unclear in this regard. Accordingly, we modified the text to soften the tone to clarify that the number of sites of our study is insufficient to predict the efficiency of each enzyme. We are also **willing to further edit** any mention of the prediction of editing efficiency that you may consider necessary. With that, we hope that you'll find our present findings and conclusions of value to be published in Nature Communication as the other Referee did. We are convinced that adding many more PAM sites is unrealistic for us and beyond the scope of this study. In any case, we would like to thank you again for your valuable time in reviewing our manuscript.